# VL-Cache: Sparsity and Modality-Aware KV Cache Compression for Vision-Language Model Inference Acceleration

**Dezhan Tu** [*]
Department of Electrical and Computer Engineering
University of California, Los Angeles
Los Angeles, CA, USA
dztu@g.ucla.edu

**Danylo Vashchilenko**
AWS AI
Amazon
New York, NY, USA
vdanylo@amazon.com

**Yuzhe Lu**
AWS AI
Amazon
Santa Clara, CA, USA
yuzhelu@amazon.com

**Panpan Xu**
AWS AI
Amazon
Santa Clara, CA, USA
xupanpan@amazon.com

## Abstract

Vision-Language Models (VLMs) have demonstrated impressive performance across a versatile set of tasks. A key challenge in accelerating VLMs is storing and accessing the large Key-Value (KV) cache that encodes long visual contexts, such as images or videos. While existing KV cache compression methods are effective for Large Language Models (LLMs), directly migrating them to VLMs yields suboptimal accuracy and speedup. To bridge the gap, we propose VL-Cache, a novel KV cache compression recipe tailored for accelerating VLM inference. In this paper, we first investigate the unique sparsity pattern of VLM attention by distinguishing visual and text tokens in prefill and decoding phases. Based on these observations, we introduce a *layer-adaptive sparsity-aware cache budget allocation* method that effectively distributes the limited cache budget across different layers, further reducing KV cache size without compromising accuracy. Additionally, we develop a *modality-aware token scoring policy* to better evaluate the token importance. Empirical results on multiple benchmark datasets demonstrate that retaining only 10% of KV cache achieves accuracy comparable to that with full cache. In a speed benchmark, our method accelerates end-to-end latency of generating 100 tokens by up to 2.33x and speeds up decoding by up to 7.08x, while reducing the memory footprint of KV cache in GPU by 90%.

## 1 Introduction

Vision-Language Models (VLMs) have recently emerged as powerful tools for a broad range of multi-modal tasks (Liu et al., 2023b; Chen et al., 2023; Bai et al., 2023). As these models improve at processing long visual context – such as high-resolution images, multiple images and multi-frame videos (Li et al., 2024a) – the number of visual tokens increase rapidly. Consequently, deploying VLMs demands substantial GPU memory capacity, bandwidth, and computational resources, leading to high inference latency and cost.

Similarly to Large Language Models (LLMs) (Chang et al., 2024), VLMs decode tokens sequentially in an auto-regressive loop. The key and value pairs of the input prompt and of the generated output tokens are stored in GPU memory (where they are known as the KV cache) and reused at each decoding step to avoid recomputation. As the context length grows, KV cache not only occupies a larger amount of GPU memory, but also increases inference latency due to data movement between GPU's high-bandwidth memory (HBM) and its on-chip memory (SRAM) in each decoding

---

[*]Work done during internship at Amazon AWS AI

step (Dao, 2023; Hong et al., 2024). This is a significant challenge for scaling VLMs, because large KV cache is required to hold the input images and video frames. For example, with a batch size of four prompts, five images in each prompt, and each image using 2K visual tokens, serving the LLaVA-1.6-34B model requires 110 GB of HBM capacity just for the KV cache of visual tokens.

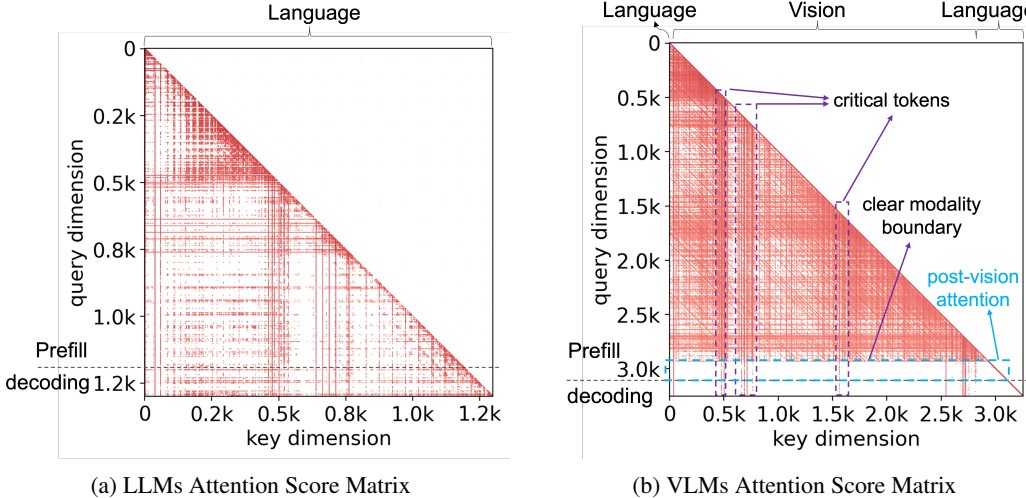

(a) LLMs Attention Score Matrix        (b) VLMs Attention Score Matrix

Figure 1: Attention Score Matrix from LLaVA v1.6 Mistral 7B with (a) language-only context, and (b) language and vision context. A deeper red color indicates a higher attention score. Both matrices indicate that critical tokens used in the decoding phase are primarily consistent with those in the prefill context. The key difference is that, in (b) VLMs attention, a clear modality boundary emerges along the query dimension.

KV cache sparsification is a promising method of reducing GPU memory requirements and inference latency for LLMs (Zhang et al., 2024b;a; Yang et al., 2024a; He et al., 2024; Li et al., 2024b). Prior methods leveraged sparsity in the attention scores matrix of LLMs to evict insignificant tokens from KV cache, while preserving important contextual information needed for output token generation. Besides reducing the GPU memory footprint, this method also reduces the token generation latency by minimizing data movements and floating-point operations (FLOPs) in the attention layer.

We discover that VLMs and LLMs exhibit significantly different attention sparsity patterns, as illustrated in Figure 1. In VLMs, a clear modality boundary exists between visual tokens and the subsequent language tokens. Specifically, the attention patterns from the output language tokens is much closer aligned with the language tokens that follow the visual tokens in the prompt (the **post-vision attention**) rather than the visual tokens themselves. Formally, we define **post-vision attention** as the sub attention score matrix that's sliced along the query dimension to only include language prompt tokens after vision tokens, as illustrated in Figure 1(b). Previous KV compression methods are modality-unaware and incorrectly conflate these two sources of attention scores. Unsurprisingly, our experiment in Figure 9 shows that KV cache compression methods designed for LLMs have suboptimal results when applied to VLMs.

Next, we measure that attention sparsity ratios vary in 70% to 99% range across transformer layers. Our observations in Section 3.1 show that previous KV compression methods suboptimally distribute the cache budget between layers, which leads to either under- or over-compression of information in KV cache. Finally, we observe that sparsity ratios differ between visual and language tokens as well, so an optimal cache budget allocation between layers can not be done before sparsity is measured in a particular prompt.

To address these gaps, we propose VL-Cache, a novel KV cache compression method for accelerating **V**ision-**L**anguage Model inference. Our method fully utilizes cross-modality and layer-wise attention sparsity patterns in VLMs to dynamically prune KV cache with minimal loss of task-level accuracy. To the best of our knowledge, this is the first work that investigates attention sparsity in VLMs and the first work that specifically optimizes KV cache compression for VLMs. In particular, we propose (1) sparsity-aware KV cache budget allocation between the transformer layers at inference time, and (2) modality-aware scoring policy for token eviction.

Experimental results in Section 5.1 show that our method retains 98% of the original task-level accuracy while using only 10% of KV cache for the majority of vision-language tasks from several datasets. In a speed benchmark, our method reduces end-to-end latency of generating 100 tokens by up to **2.33x** and speeds up decoding in particular by up to **7.08x**, while allocating **90% less** GPU memory for KV cache. In inference scenarios where KV cache size is the limiting factor to higher concurrency, VL-Cache enables up to **10x** higher concurrency after KV cache compression.

Overall, our contributions in this paper are summarized as follows:

- **VLMs Attention Sparsity Profile.** We uncovered unique attention sparsity patterns in the prefill and decoding phases of VLM inference on a variety of multi-modal tasks, which is drastically different from those of LLMs.
- **Layer-Adaptive Sparsity-Aware Cache Budget Allocation.** We proposed to allocate each layer's KV cache budget with consideration of that layer's attention sparsity at inference time. Even with 10% KV cache budget, we retained high accuracy on popular benchmark tasks.
- **Modality-aware Token Scoring Policy.** We observed that language-to-vision attention robustly includes information about the importance of visual tokens. We treat visual and language attention scores differently to better retain important tokens.

## 2 BACKGROUND

In this section, we detail the inference procedure of the widely adopted VLM architecture with vision & language input and text output, where image tokens are projected as soft prompts (Li et al., 2024a; Team et al., 2023; Islam & Moushi, 2024; Bai et al., 2023; Anthropic, 2023). We also introduce the formulation of KV cache compression and the approach of existing works (Zhang et al., 2024b;a; Liu et al., 2024a; Ge et al., 2023; Tang et al., 2024; Li et al., 2024b; Lee et al., 2024; Yu et al., 2024).

### 2.1 VLM INFERENCE

**Prefill phase.** The input to VLMs includes both images and language, where images are processed by the visual encoder to generate visual tokens. Subsequently, a projection layer, such as a simple multi-layer perceptron, maps these visual tokens to a unified embedding space. Meanwhile the language prompt is fed to a tokenizer and embedding layer to create the initial hidden state for language tokens. For notation simplicity, we denote a sequence of $m$ prompt tokens, including both visual and language ones, as $\{x_1, ..., x_m\}$. These tokens are processed by the language model in parallel to calculate the probability of the first decoded token $P_\theta(x_{m+1}|x_1, ..., x_m)$. Simultaneously, the key vectors $\{k_1^{(l)}, ..., k_m^{(l)}\}$ and value vectors $\{v_1^{(l)}, ..., v_m^{(l)}\}$ at each transformer layer $l$ are cached in GPU memory to avoid recomputation in the next decoding step.

**Decoding phase.** Once decoding starts, the language model in VLMs takes effect and generates one token per step in an auto-regressive loop. At step $i$, the model receives the token $x_{m+i}$ as input and calculates the probability $P_\theta(x_{m+i+1}|x_1, ..., x_{m+i})$. Each decoding step involves generating new key vectors $k_{m+i}^{(l)}$ and value vectors $v_{m+i}^{(l)}$, which are appended to the previously cached key-value pairs for each layer, resulting in $\{k_1^{(l)}, ..., k_{m+i}^{(l)}\}$ and $\{v_1^{(l)}, ..., v_{m+i}^{(l)}\}$. In case of long contexts, such as multiple or high resolution images, the key-value cache can grow significantly larger than the model parameters and other intermediate tensors, making memory capacity and bandwidth major performance bottlenecks.

### 2.2 KV CACHE COMPRESSION

To address the bottleneck of storing and accessing the large KV cache during decoding, many researchers have focused on KV cache compression to maintain only a subset of the full KV cache for more efficient decoding while minimizing the accuracy loss. There are two main design dimensions to such algorithms: how many cache tokens should be kept at each layer, and which tokens to evict during compression.

**Budget Allocation.** Since the transformer architecture consists of multiple identical layers, a straightforward strategy is to allocate an equal budget of KV cache slots to each layer (Xiao et al.,

2023b; Zhang et al., 2024b; He et al., 2024). More recently, inspired by the observation that removing cache tokens at different layers results in varying degrees of performance loss, PyramidKV (Zhang et al., 2024a) and PyramidInfer (Yang et al., 2024a) proposed a decay schedule to assign more cache budget to shallower layers and observed improved accuracy results relative to the baseline with equal budget per layer.

**Token Scoring Policy.** For a given cache token budget, a token scoring policy $\psi$ is required to rank the importance of KV cache tokens and decide which tokens to keep. Let $n$ be the count of cache tokens at layer $l$, and $S = \{0, 1, ..., n\}$ be the indices of these cache tokens. We define $\psi : S \to \mathbb{R}^n$, with the output being the scores for current cache tokens. Given these scores, indices with top-$k$ scores, $S_\psi := \{i_1, i_2, ..., i_k : \psi(S)_{i_j} \geq \psi(S)_{x \in [n] \setminus \{i_1, i_2, ..., i_k\}}\}$, where $k \in [1, n)$, are selected. Recent works have shown that attention scores serve as an effective source to design such policies. StreamingLLM (Xiao et al., 2023b) found that high attention scores are frequently assigned to initial tokens and achieved length generalization by keeping only the initial and most recent cache tokens while removing intermediate ones. H2O (Zhang et al., 2024b) uses accumulated attention scores to identify crucial tokens to retain.

We make the observation that, fundamentally, KV cache compression methods only work because inference with a transformer layer is a sparse process. Therefore, in this work, we leverage attention sparsity as the unified guidance to design both the cache budget allocation mechanism and the token eviction policy for KV cache compression in VLMs.

## 3 PRELIMINARY EXPERIMENT

The attention mechanism for visual and language tokens is a key aspect of VLMs. Therefore, further optimization of the KV cache compression methods for VLMs requires careful investigation of the attention patterns with relevant input prompts. Motivated by this need, we conducted preliminary experiments to explore the VLM attention. We randomly sampled data from three multi-modal datasets — DocVQA (Mathew et al., 2021), MathVista (Lu et al., 2023), and Coco-Caption (Chen et al., 2015). These datasets cover of a wide range of visual tasks, including OCR, visual diagram reasoning, and world knowledge. We selected one of the state-of-the-art VLMs, LLaVA-Mistral-7B (Liu et al., 2023b), and recorded the attention score matrix until the generation process completes. We will leverage insights from these analyses to motivate our algorithm design in Section 4.

### 3.1 MEASURING ATTENTION SPARSITY

In this section, we measure the sparsity of the attention score matrix in different transformer layers during the prefill and decoding phases. First, we apply a filter with a relative threshold $p$ to the attention score matrix $A$:

$$\textbf{ThresholdFilter}(A, p)_{ij} = \begin{cases} A_{ij} & \text{if } A_{ij} \geq p \cdot \max_j(A_{ij}) \\ 0 & \text{otherwise} \end{cases} \tag{1}$$

where threshold $p \in (0, 1)$ controls the strength of induced sparsification, following Zhang et al. (2024b). We also heuristically set $p = 1\%$, such that the filtered-out scores have little impact on the output of the transformer layer. After filtration, we calculate sparsity $\gamma^{(l)} \in [0, 1]$ of layer $l$ as count of zero entries, normalized by the size of the lower triangular portion of the attention scores matrix:

$$\gamma^{(l)} := \frac{\sum_{i \geq j} \mathbb{1}[\textbf{ThresholdFilter}(A^{(l)}, p)_{ij} = 0]}{|\{A_{ij}^{(l)} : i \geq j\}|} \tag{2}$$

We calculated the average sparsity over a sample from the evaluation datasets, and then plotted the range of each layer's sparsity with different attention heads in Figure 2. We observe that in the prefill phase (Figure 2a), the first two layers exhibit significantly lower sparsity and higher density compared to other layers. Additionally, some layers in the middle also demonstrate higher density than their neighboring layers. During the decoding phase (Figure 2b), a similar trend is observed. Based on these observations, we expect that the aggregate attention sparsity during prefill could effectively predict the required KV cache size for robust decoding.

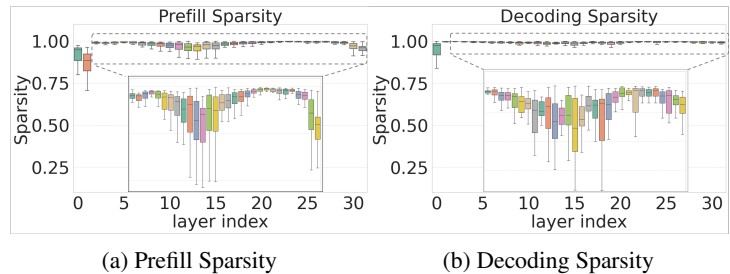
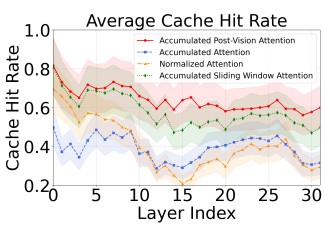

| (a) Prefill Sparsity | (b) Decoding Sparsity |
|---|---|

Figure 2: Layer-wise attention sparsity in prefill and decoding phases. Different layers exhibit varying degrees of sparsity; The layer-wise sparsity trend in the decoding phase is similar to that in the prefill phase.

Figure 3: Average Cache Hit Rate. The accumulated post-vision attention (ours) demonstrates a higher cache hit rate compared to the other two token scoring policies.

We note that previous methods, such as H2O (Zhang et al., 2024b) and Keyformer (Adnan et al., 2024), allocate the same cache size across all layers, leading to insufficient allocation for high-information layers and wasteful allocation for low-information layers. More recent work including PyramidKV (Zhang et al., 2024a) and PyramidInfer (Yang et al., 2024a) monotonically decrease the KV cache size with depth of each layer, which we find to be suboptimal as well. Our observations reveal a more nuanced sparsity pattern, where different layers demand non-monotonically varying cache sizes to retain context information during decoding.

### 3.2 MEASURING CACHE HIT RATE

In Figure 1b of Section 1, we previously noted a distinct boundary in VLM attention between vision and language tokens in the prompt. Based on these observations, we further hypothesize that retaining tokens based on post-vision prefill attention instead of the full-prompt prefill attention would preserve important cache tokens with higher recall. To validate this hypothesis, we define **CacheHitRate** to measure the fraction of important tokens that are preserved after eviction when different scoring policies are applied. Let $m$ be the number of prompt tokens, $Q_{:m}, K_{:m} \in \mathbb{R}^{m \times d}$ be the corresponding query and key matrices, and $Q_{m+1} \in \mathbb{R}^d$ be the query vector of the first decoding token. Also recall from Section 2.2 that $S$ denotes the indices of prompt cache tokens, and $\psi$ denotes a scoring policy that maps each token index to a real number as their importance.

**Definition 3.1** (CacheHitRate), given $A_{m+1} := softmax(\frac{Q_{m+1}K_{:m}{}^T}{\sqrt{d}}) \in \mathbb{R}^m$, we define:

- $\psi^* : S \to A_{m+1}$ as the optimal scoring function since $A_{m+1}$ is true attention scores in decoding;
- $S_{\psi^*,k}$ as the top-$k$ tokens selected per $\psi^*$'s ranking, which we treat as the ground-truth for all $k$;
- **CacheHitRate** as $\frac{|S_{\psi,k} \cap S_{\psi^*,k}|}{|S_{\psi^*,k}|}$, the percentage of true top-$k$ tokens $S_{\psi^*,k}$ that is also preserved by $S_{\psi,k}$, which are top-$k$ tokens kept by any policy $\psi$.

We note that during inference, $A_{m+1}$ is not available because we aim to compress KV cache before the decoding pass to lift the memory bottleneck and boost decoding throughput. Thus, multiple $\psi$s have been proposed to approximate $\psi^*$ in recent works. In the following paragraphs, we use $\psi(S)$ to indicate the token scores assigned to $S$ with an arbitrary policy $\psi$. For brevity, we assume the $softmax$ outputs below are applied necessary causal masks to serve as attention scores.

*Accumulated Attention* (prior work) uses the cumulative attention score along the query dimension, $\psi(S) := \sum_i softmax(\frac{Q_{:m}K_{:m}{}^T}{\sqrt{d}})_i$ to rank the importance of cache tokens.

*Normalized Attention (prior work)* instead computes the average attention scores along the query dimension based on the observation that accumulated scores disproportionately favors earlier tokens. The policy can be written as $\psi(S) := \frac{1}{\mathbf{n}} \odot \sum_i softmax(\frac{Q_{:m}K_{:m}{}^T}{\sqrt{d}})_i$, where vector $\mathbf{n}$ denotes the number of unmasked elements in each column and $\odot$ indicates element-wise multiplication.

*Sliding Window Attention* (prior work) also computes accumulated attention scores along the query dimension but only over a recent window. This policy is defined as $\psi(S) :=$

$\sum_i softmax(\frac{Q_{m-w:m}K_{m-w:m}^T}{\sqrt{d}})_i$, where $w$ is a fixed window size. Since $w$ is fixed, whether the score is normalized does not change the ranking of cache tokens for evicting purposes.

**Post-vision Attention** (ours) is based on the observation of the similarity between attention from post-vision prompt tokens and from decoding tokens as shown in Figure 1b. We adopt the term post-vision to distinguish language tokens that follow visual tokens from instructions that precede the images in the prompt. Formally, we use the portion of the attention scores $Q_{m-\tau:m}$, where $\tau$ is the count of language tokens that follow the vision tokens in the prompt. Our policy is defined by $\psi(S) := \sum_i softmax(\frac{Q_{m-\tau:m}K_{m-\tau:m}^T}{\sqrt{d}})_i$. During prefill, one can interpret our policy as a dynamic prompt-dependent sliding window with window size set to the length of the post-vision language prompt instead of a static prompt-independent value. From Figure 3, we find our scoring policy based on post-vision attention leads to consistently higher cache hit rate across layers than other policies.

## 4 VL-CACHE METHOD

Motivated by observations from preliminary experiments, we introduce our method VL-Cache, which strategically combines sparsity-aware cache budget allocation and modality-aware token scoring policy to improve VLM's performance under limited KV cache budget, in terms of both accuracy and efficiency.

Specifically, we use Post-vision Attention to compute both inter-layer sparsity and intra-layer token importance. The former guides how many cache tokens should be allocated at each layer, while the latter dictates which $k$ tokens within a layer should be kept due to their importance. A high-level description of our method is visualized in Figure 4. For brevity, we will use $A'$ to denote the Post-vision Attention matrix in this section.

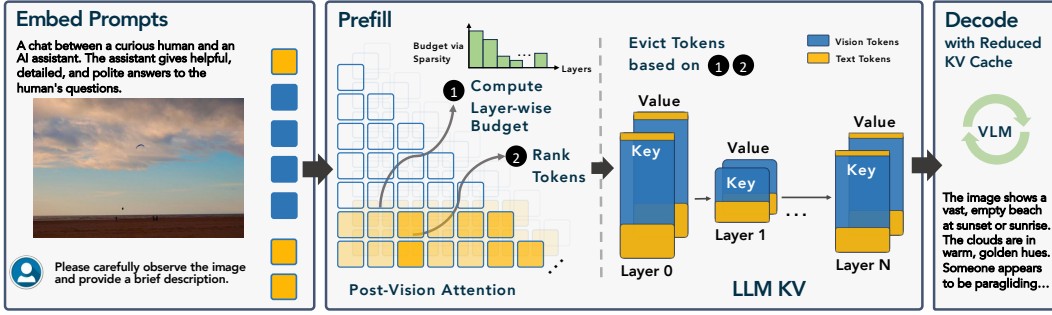

Figure 4: VL-Cache Overview. During prefill, post-vision attention matrices are utilized to compute sparsity-driven layer-wise budgets and rank the importance of cache tokens. Unimportant tokens are then evicted, allowing lower memory usage and accelerated decoding with reduced KV cache.

We argue that using Post-vision Attention has two important advantages. Firstly, computing layer-wise sparsity using Post-vision Attention results in much lower memory and latency $O(\tau m)$ compared to using the full attention matrix $O(m^2)$ as in (Zhang et al., 2024b) and (He et al., 2024), since the visual tokens dominate the prompt length, $\tau \ll m$ for current vision-language tasks. Second, with a fixed cache budget, using Post-vision Attention leads to better preservation of important tokens, as measured by high **CacheHitRate** in Figure 3. In this section, we will detail how we perform sparsity-aware KV cache budget allocation and cache token eviction guided by Post-vision Attention before reviewing the strong performance of VL-Cache in Section 5.

### 4.1 SPARSITY-AWARE KV CACHE BUDGET ALLOCATION

Before determining the exact tokens to evict, we need to allocate the KV cache budget, which is the percentage of the KV cache to retain at each layer. Based on our observations from Section 3.1, we implement a *sparsity-aware layer-wise KV cache allocation* approach with two steps during the prefill phase, detailed in Algorithm 1 below. First, we apply **ThresholdFilter** pruning (with $p = 1\%$) to the Post-vision Attention scores and calculate the layer-wise sparsity (line 4-8 and 13-18). Second, given a target KV cache budget for the whole model, we distribute this budget across

layers based on each layer's sparsity ratio (line 9-12). This method optimizes the use of limited memory to store an appropriate amount of context information in each layer. Since the allocation occurs only once right after prefill, the latency overhead is amortized across multiple decoding steps.

---

**Algorithm 1** Sparsity-Aware Cache Budget Allocation

---

1: **Input:** query and key $Q, K \in \mathbb{R}^{L \times H \times m \times d}$, number of layers $L$, number of heads $H$, length of post-vision prompt $\tau$, overall cache budget $\alpha$.
2: **procedure** SKEWEDCACHEBUDGETALLOCATION($Q, K, \alpha, L, H$)
3: $\quad \triangleright$ *main method to compute layer-wise cache budget.* $\quad\quad\quad\quad\quad\quad\quad\quad\quad\quad\quad\quad\quad\quad \triangleleft$
4: $\quad \Gamma[L][H] \leftarrow 0$ $\quad\quad\quad\quad\quad\quad\quad\quad\quad\quad\quad\quad\quad \triangleright \Gamma$ *stores layer and head-wise sparsity*
5: $\quad$ **for** $l = 1 \rightarrow L$ **do**
6: $\quad\quad$ **for** $h = 1 \rightarrow H$ **do**
7: $\quad\quad\quad \Gamma_h^{(l)} \leftarrow$ ComputePostVisionSparsity($Q_h^{(l)}, K_h^{(l)}$) $\quad \triangleright$ *invoke helper method below*
8: $\quad \gamma \leftarrow \Gamma.\text{mean}(1)$ $\quad\quad\quad\quad\quad\quad\quad\quad\quad \triangleright \gamma$ *is head-averaged sparsity for each layer*
9: $\quad Z \leftarrow \sum_l 1 - \gamma^{(l)}$ $\quad\quad\quad \triangleright Z$ *is the sum of non-sparse ratios across layers as the normalizer*
10: $\quad \beta[L] \leftarrow 0$ $\quad\quad\quad\quad\quad\quad\quad\quad\quad\quad\quad \triangleright \beta$ *stores cache token budget for each layer*
11: $\quad$ **for** $l = 1 \rightarrow L$ **do**
12: $\quad\quad \beta^{(l)} \leftarrow \text{clip}(\frac{1.0 - \gamma^{(l)}}{Z} \alpha L, 0.01, 1)$ $\quad \triangleright$ *allocate budget by normalized non-sparse ratio*
13: **procedure** COMPUTEPOSTVISIONSPARSITY($Q, K$)
14: $\quad \triangleright$ *helper method to compute post-vision attention head sparsity.* $\quad\quad\quad\quad\quad\quad\quad \triangleleft$
15: $\quad Q' \leftarrow Q_{m-\tau:m}$ $\quad\quad\quad\quad\quad\quad\quad\quad\quad\quad\quad \triangleright Q'$ *is post-vision query*
16: $\quad A' \leftarrow softmax(\frac{Q'K^T}{\sqrt{d}})$ $\quad\quad\quad\quad\quad\quad\quad\quad \triangleright A'$ *is post-vision attention*
17: $\quad \gamma' \leftarrow \sum_{i+m-\tau \geq j} \frac{\mathbb{1}[\textbf{ThresholdFilter}(A', p)_{ij} = 0]}{|\{A'_{ij} : i + m - \tau \geq j\}|}$ $\quad \triangleright \gamma'$ *is ratio of zeros in unmasked entries*
18: $\quad$ **return** $\gamma'$
19: **Output:** layer-wise budget $\beta$

---

In Algorithm 1, we present our KV cache budget allocation algorithm, where $\alpha$ is a hyper-parameter that encodes the desired KV cache budget for the whole model. When the accuracy is not satisfactory, a higher $\alpha$ can be used to keep more KV cache. Finally, we would like to point out the key difference between our algorithm and PyramidKV (Zhang et al., 2024a): our allocation budget is customized for each prompt based on its sparsity pattern, instead of using a fixed layer-wise budget for all prompts, thus granting additional flexibility.

## 4.2 MODALITY-AWARE TOKEN SCORING POLICY

After we decide the cache budget for layer $l$, we choose a subset of $k^{(l)}$ cache tokens from the full cache. Prior works (that were targeting KV cache compression for LLMs) have explored several scoring policies, such as *Accumulated Attention* in H2O (Zhang et al., 2024b), *Normalized Attention* in ZipCache He et al. (2024) and *Sliding Window Attention* in Li et al. (2024b); Zhang et al. (2024a); Yang et al. (2024a), as we discussed earlier in Section 3.2. In this section, we will explain in-depth why *Post-vision Attention* provides a better scoring policy than prior policies for VLMs.

While *Accumulated Attention* serves as a simple yet effective baseline, one critical issue is that summation over the full query dimension unavoidably assigns high scores to earlier tokens. Observing this length bias, *Sliding Window Attention* only sums attention scores over a recent window. We confirm that this indeed leads to higher **CacheHitRate**, as shown in Figure 3. However, as we can see in Figure 1a and 1b, VLMs have a significantly different attention pattern than LLMs. Along the query dimension, we observe a clear modality boundary: visual tokens attend to other visual tokens rather uniformly while language tokens only pay attention to a few visual tokens with high concentration. If we were to apply *Accumulated or Normalized* Attention as the scoring policy, the signal for critical cache tokens will be buried as we sum or average scores from all the preceding visual tokens. Using a recent window helps, but a fixed window size can easily be too large or too small as the length of the post-vision prompt varies from case to case. Therefore, we introduce *Post-vision Attention* as an optimized scoring policy for VLMs by implementing a dynamic, prompt-specific window size. The advantage of our scoring policy is evident by its high **CacheHitRate** in Figure 3, and will be further demonstrated through accuracy comparisons in Section 5.

# 5 EXPERIMENTS

In our experiments, we evaluate VL-Cache across representative VLMs that can handle image, videos, and language inputs. We use the state-of-the-art open-source LLaVA family, including LLaVA-Mistral-7B (Liu et al., 2023b) and LLaVA-1.6-34B (Liu et al., 2023a). They all share the same visual model (openai/clip-vit-large-patch14-336 (Radford et al., 2021)) but are fine-tuned from different language backbones (e.g., Mistral (Jiang et al., 2023), Nous-Hermes-2-Yi-34B (Research, 2023)). Also, the former model uses Grouped Query Attention (GQA) (Ainslie et al., 2023), while the latter model uses Multi-Head Attention (MHA) (Vaswani et al., 2017).

**Implementation Details**. We use an AWS EC2 P4 instance equipped with 8 A100 40GB GPUs for evaluation. First, we sample three tasks from lmms-eval (Bo Li* & Liu, 2024), including Coco-Caption (Chen et al., 2015), DocVQA (Mathew et al., 2021), and MathVista (Lu et al., 2023). These tasks are representative, and cover OCR, reasoning, and world knowledge domains. Second, we compare accuracy of our approach against full-cache baselines and previous KV cache sparsification methods including StreamingLLM (Xiao et al., 2023a), H2O (Zhang et al., 2024b), ZipCache (He et al., 2024) and PyramidKV (Zhang et al., 2024a). In appendix A.3, we show more comprehensive experimental results with additional datasets (TextVQA (Singh et al., 2019), ChartQA (Masry et al., 2022)) and methods that do not focus on KV cache compression but are still relevant (FastV (Chen et al., 2025) and HiRED (Arif et al., 2024)). We apply KV cache sparsification in line with these baselines by retaining the most recent tokens and selecting the Top-K tokens according to their corresponding scoring policies. All baselines are configured with their default settings, except that the KV cache budget is scaled proportionally to the prompt length, and the recent token window size is fixed at 10% of this budget to enable a fair comparison. Finally, we benchmark latency with varying context lengths and batch sizes.

## 5.1 ACCURACY EVALUATION

The accuracy evaluation results are shown in Figure 9 and Table 3. We report the average accuracy score with KV cache budget varying from 1% to 100% of prompt length. Overall, VL-Cache outperforms other baselines across a range of KV cache budgets and different language model backbones.

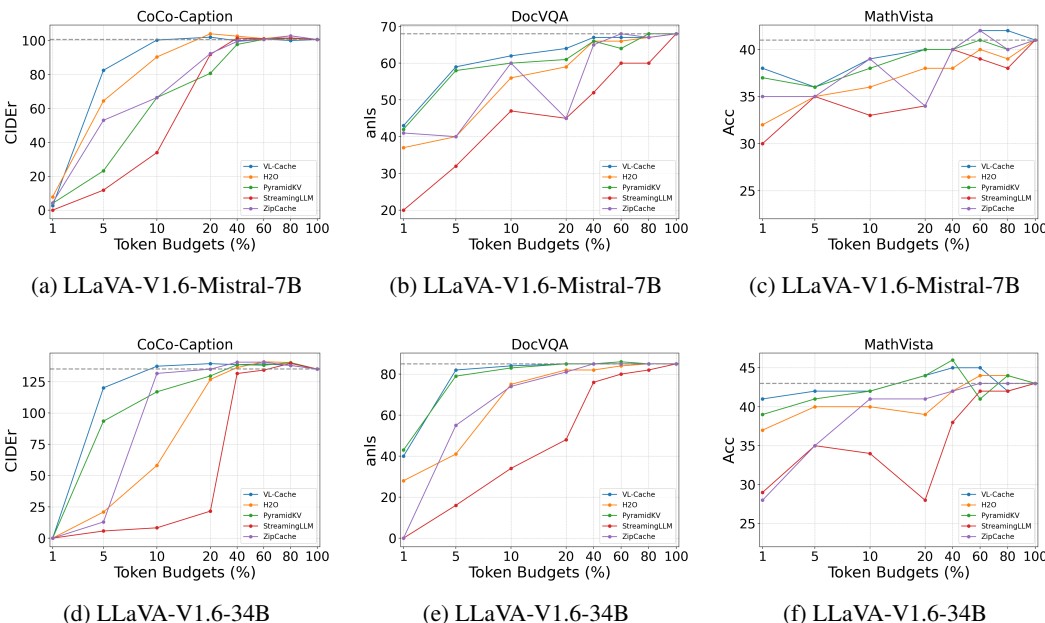

Figure 5: Evaluation results on different datasets with varied cache budgets. VL-Cache achieves comparable accuracy against full-cache and outperforms multiple baselines with limited KV cache budget. Interestingly, we found that VLMs occasionally perform slightly better with a partial KV cache. We attribute it to the regularization effect of KV cache compression.

For the Coco-Caption dataset (Figure 9 (a), (c) and Table 3), all baselines maintain high accuracy when the KV cache budget exceeds 40% for VLMs. However, as the KV cache budget is further reduced, accuracy significantly declines. It is worth noting that even with only 5% to 10% KV cache budget, VL-Cache can achieve accuracy that is comparable to the full KV cache. H2O and StreamingLLM allocate the same cache budget across all layers, causing denser layers to miss important tokens in the context, while sparser layers contain redundant tokens. PyramidKV statically allocates KV cache in a monotonically decreasing manner, which is ineffective for all input queries. For the DocVQA dataset (Figure 9 (b), (d) and Table 3), PyramidKV demonstrates strong performance; however, VL-Cache consistently outperforms all other baselines. H2O and StreamingLLM lack robustness, with performance significantly declining as the KV cache is further compressed. For the MathVista dataset (Figure 9 (c), (f) and Table 3), our method again outperforms other methods in most cases. Meanwhile, multiple methods achieve near full-cache performance with 1% KV cache. It's likely due to the fact that 54% of the dataset are multiple-choice questions, which are less challenging than free-form questions.

| Dataset | Model | Method | 1% | 5% | 10% | 20% | 40% | 60% | 80% | 100% |
|---|---|---|---|---|---|---|---|---|---|---|
| Coco-Caption (Metric: CIDEr) | LLaVA-Mistral-7B | VL-Cache | 2.64 | **82.53** | **100.36** | 102.06 | 99.93 | 101.07 | 100.08 | 100.68 |
| | | H2O | **7.87** | 64.45 | 90.36 | **104.04** | **102.64** | **101.21** | **102.86** | 100.68 |
| | | PyramidKV | 4.01 | 23.21 | 66.41 | 80.76 | 97.76 | 100.75 | 101.38 | 100.68 |
| | | StreamingLLM | 0.05 | 11.82 | 33.98 | 91.87 | 101.47 | 101.07 | 101.6 | 100.68 |
| | | ZipCache | 4.4 | 53.00 | 66.41 | 92.36 | 99.51 | 100.71 | **102.86** | 100.68 |
| | LLaVA-1.6-34B | VL-Cache | 0 | **120.11** | **137.35** | **139.42** | 138.58 | 139.19 | 138.01 | 135.07 |
| | | H2O | 0 | 20.87 | 58.14 | 126.8 | 136.27 | **140.89** | **140.30** | 135.07 |
| | | PyramidKV | 0 | 93.47 | 116.91 | 129.59 | 138.53 | 138.17 | 140.15 | 135.07 |
| | | StreamingLLM | 0 | 5.69 | 8.23 | 21.57 | 131.51 | 134.26 | 139.65 | 135.07 |
| | | ZipCache | 0 | 12.92 | 131.61 | 134.98 | **140.59** | 140.63 | 137.90 | 135.07 |
| DocVQA (Metric: ANLS) | LLaVA-Mistral-7B | VL-Cache | **43** | **59** | **62** | **64** | **67** | 67 | 67 | 68 |
| | | H2O | 37 | 40 | 56 | 59 | 66 | 66 | 67 | 68 |
| | | PyramidKV | 42 | 58 | 60 | 61 | 66 | 64 | **68** | 68 |
| | | StreamingLLM | 20 | 32 | 47 | 45 | 52 | 60 | 60 | 68 |
| | | ZipCache | 41 | 40 | 60 | 45 | 65 | **68** | 67 | 68 |
| | LLaVA-1.6-34B | VL-Cache | 40 | **82** | **84** | **85** | **85** | 85 | **85** | 85 |
| | | H2O | 28 | 41 | 75 | 82 | 82 | 84 | **85** | 85 |
| | | PyramidKV | **43** | 79 | 83 | **85** | **85** | **86** | **85** | 85 |
| | | StreamingLLM | 0 | 16 | 34 | 48 | 76 | 80 | 82 | 85 |
| | | ZipCache | 0 | 55 | 74 | 81 | **85** | 85 | **85** | 85 |
| MathVista (Metric: ACC) | LLaVA-Mistral-7B | VL-Cache | **38** | **36** | **39** | **40** | **40** | **42** | **42** | 41 |
| | | H2O | 32 | 35 | 36 | 38 | 38 | 40 | 39 | 41 |
| | | PyramidKV | 37 | **36** | 38 | **40** | **40** | 41 | 40 | 41 |
| | | StreamingLLM | 30 | 35 | 33 | 34 | **40** | 39 | 38 | 41 |
| | | ZipCache | 35 | 35 | 39 | 34 | **40** | **42** | 40 | 41 |
| | LLaVA-1.6-34B | VL-Cache | **41** | **42** | 42 | 44 | 45 | **45** | 42 | 43 |
| | | H2O | 37 | 40 | 40 | 39 | 42 | 44 | **44** | 43 |
| | | PyramidKV | 39 | 41 | 42 | 44 | **46** | 41 | **44** | 43 |
| | | StreamingLLM | 29 | 35 | 34 | 28 | 38 | 42 | 42 | 43 |
| | | ZipCache | 28 | 35 | 41 | 41 | 42 | 43 | 43 | 43 |

Table 1: With 10% KV cache, VL-Cache achieves close to 100% of full cache performance on most datasets and models and significantly outperformed other methods. Since all KV cache compression methods will converge to full cache performance as the more cache budget is granted, strong performances at low cache budgets attest to VL-Cache's effectiveness in retaining important KV tokens.

## 5.2 SPEED BENCHMARK

In order to show the speed advantage of VL-Cache, we measure the GPU kernel latencies of prefill and decoding forward passes with synthetic prompts, following the method in Kwon et al. (2023). We vary the size of the prompt from 1K tokens to 128K tokens to scale our method to a very long context. Batch sizes vary from 1 to 64 and are static, meaning that all requests get prefilled and decoded concurrently. We assume that the prompt template format remains similar to our accuracy benchmarks, so we use the last 50 tokens of the prompt to determine which tokens to evict from the KV cache. For both prefill and decoding in the baseline, we used default settings from the HuggingFace implementation [1], including CUDA-based FlashAttention-v2. To optimize performance in our VL-Cache, we applied our Triton-based solution for self-attention forward pass, layer-wise

---
[1]https://huggingface.co/docs/transformers/main/en/perf_infer_gpu_one

sparsity evaluation, and modality-aware token scoring, as detailed in appendix A.5. The speedup is calculated as $\frac{\text{Baseline latency}}{\text{VLCache latency}}$.

In Table 2, we observe that with 50 query tokens for calculating attention statistics, the overhead of our method is just 1-4% of the prefill latency. See Appendix A.6 for detailed measurements of the overhead and a discussion on how to reduce the overhead of statistics calculation for a large number of query tokens. During decoding, we run 99 forward passes for a total of 100 output tokens. We observe up to 7x decoding speedups that are attributed to the reduced size of the KV cache, which we compressed to 10% for this benchmark.

Overall, we see that the end-to-end speedup is bounded by the prefill latency, which is mostly unchanged by VL-Cache. For example, with prompt length of 128K and batch size of 1, the decoding speedup of 7.08x is diluted by prefill taking 53% of end-to-end latency, which results in 1.66x end-to-end speedup. End-to-end speedup will monotonically approach the decoding speedup as the count of output tokens increases. This highlights that KV cache sparsity will give the best speed advantage in tasks with long outputs, such as image captions, video descriptions, chain-of-thought multi-modal reasoning, etc.

| Batch Size | Prompt Length | Prefill Speedup | Decoding Speedup | End-to-End Speedup |
|---|---|---|---|---|
| 1 | 2k | 0.96 | 1.19 | 1.16 |
| 1 | 8k | 0.97 | 1.70 | 1.49 |
| 1 | 32k | 0.99 | 3.32 | 1.85 |
| 1 | 128k | 0.99 | **7.08** | 1.66 |
| 4 | 2k | 0.98 | 1.68 | 1.50 |
| 4 | 8k | 0.98 | 3.16 | 1.95 |
| 4 | 32k | 0.99 | 6.07 | 2.06 |
| 16 | 2k | 0.98 | 3.03 | 1.99 |
| 16 | 8k | 0.99 | 5.61 | 2.27 |
| 64 | 2k | 0.98 | 5.23 | **2.33** |

Table 2: Performance metrics by batch size and prompt length for 100 output tokens.

Figure 6: Server-level throughput v.s. request-level latency curve (prompt length = 2K). Labeled points indicate batch size.

In our implementation of both the baseline and VL-Cache, maximum batch size is limited by peak memory usage during prefill instead of KV cache size, so compression of KV cache does not lead to higher batch size. In future work, continuous batching and chunked prefill could be used to eliminate the prefill memory bottleneck, which would expose our method's advantage in raising maximum batch size when the size of KV cache is the bottleneck to higher batch size. Finally, we summarize the trade-off between request-level latency and server-level throughput in Figure 6. We note that VL-Cache offers both higher peak throughput and lower latency for any desired server-level throughput.

## 6 CONCLUSION & FUTURE WORK

In this paper, we propose VL-Cache, a novel KV cache compression optimized for VLMs. We discovered the unique sparsity patterns of visual and language tokens throughout the prefill and decoding phases. With these observations, we introduce a modality-aware token scoring policy and sparsity-aware cache budget allocation to reduce KV cache size without accuracy loss. Empirical results on multiple benchmark datasets demonstrate that when maintaining only 10% of the KV cache, our method achieves accuracy comparable to the full KV cache and outperforms all existing methods. In a speed benchmark, our method accelerates end-to-end latency of generating 100 tokens by up to 2.33x relative to the full KV cache. As for future work, we identify two opportunities. Firstly, our method only focused on compressing prefill KV cache due to the high prompt to output token ratio (median 320x) of current vision-language tasks. Periodical compression of decoded tokens can be developed for long output tasks while striking a balance between latency overhead and memory savings. Secondly, it would be interesting to extend our method for video models since video inputs will impose even more memory pressure due to KV cache.

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

# A  APPENDIX

## A.1  VISION-LANGUAGE PROMPT TEMPLATE CONSTRUCTION

Constructing prompt templates in image-based conversations is a common practice for VLMs (Li et al., 2024a; Team et al., 2023; Islam & Moushi, 2024; Bai et al., 2023; Anthropic, 2023), as it instructs language models to generate more accurate responses. For example, as illustrated in Figure 7, the input image is processed through a visual encoder and a projection layer to generate an image embedding, represented by the <image>. For language input, beyond user input, a prompt template is employed. With the appropriate prompt template design, regardless of the original image order from user input (whether before or after language inputs), there will always be a language-based instruction or question in the post-vision position, providing a strong signal for our VL-Cache to evict insignificant visual tokens.

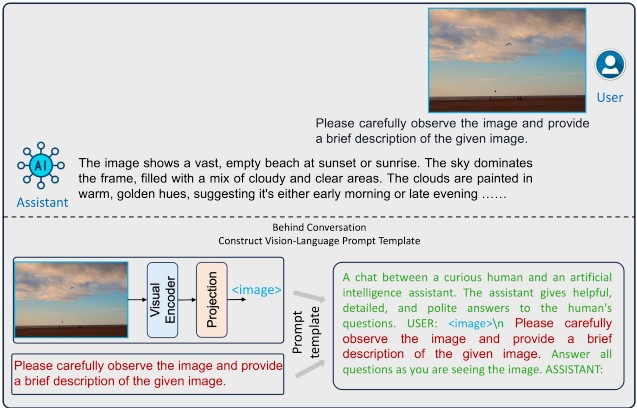

Figure 7: Vision-Language Prompt Template Example.

## A.2  ABLATION STUDIES ON THRESHOLD $p$.

In VL-Cache, we utilized **ThresholdFilter** in eq.1 to sparsify the attention matrix. It uses a relative threshold $p$ that we heuristically set to $0.01$ in our experiments. To understand the robustness of our method to this hyperparameter, we conducted an ablation study using LLaVA-V1.6-Mistral-7B and Coco-Caption. We vary $p$ from $0.0001$ to $0.1$ and show the corresponding performance in the following plot. We observe that 1) overall, VL-Cache is robust to the choice of $p$ as it has relatively small performance variations, 2) higher token budgets seem to be even more robust to $p$, and 3) there isn't a single threshold $p$ that works consistently better across all cache budgets, but $p = 0.01$ does seem to be a decent choice.

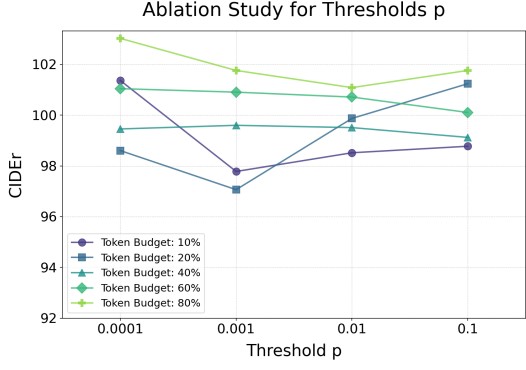

Figure 8: LLaVA-V1.6-Mistral-7B performance on Coco-Caption under various cache budgets and threshold values with VL-Cache.

## A.3 FULL EXPERIMENTAL RESULTS

| Dataset | Model | Method | 1% | 5% | 10% | 20% | 40% | 60% | 80% | 100% |
|---|---|---|---|---|---|---|---|---|---|---|
| Coco-Caption (Metric: CIDEr) | LLaVA-Mistral-7B | VL-Cache | 2.64 | **82.53** | 100.36 | 102.06 | 99.93 | 101.07 | 100.08 | 100.68 |
| | | H2O | 7.87 | 64.45 | 90.36 | 104.04 | 102.64 | 101.21 | 102.86 | 100.68 |
| | | PyramidKV | 4.01 | 23.21 | 66.41 | 80.76 | 97.76 | 100.75 | 101.38 | 100.68 |
| | | StreamingLLM | 0.05 | 11.82 | 33.98 | 91.87 | 101.47 | 101.07 | 101.6 | 100.68 |
| | | ZipCache | 4.4 | 53.00 | 66.41 | 92.36 | 99.51 | 100.71 | 102.86 | 100.68 |
| | | FastV | 9.98 | 41.17 | 88.82 | 97.47 | 105.40 | 110.60 | **106.90** | 100.68 |
| | | HiRED | **13.87** | 70.90 | **115.37** | 109.91 | 115.09 | 115.09 | 105.93 | 100.68 |
| | LLaVA-1.6-34B | VL-Cache | 0 | **120.11** | **137.35** | **139.42** | 138.58 | 139.19 | 138.01 | 135.07 |
| | | H2O | 0 | 20.87 | 58.14 | 126.8 | 136.27 | **140.89** | **140.30** | 135.07 |
| | | PyramidKV | 0 | 93.47 | 116.91 | 129.59 | 138.53 | 138.17 | 140.15 | 135.07 |
| | | StreamingLLM | 0 | 5.69 | 8.23 | 21.57 | 131.51 | 134.26 | 139.65 | 135.07 |
| | | ZipCache | 0 | 12.92 | 131.61 | 134.98 | 140.59 | 140.63 | 137.90 | 135.07 |
| | | FastV | 3.10 | 15.86 | 32.55 | 80.04 | 112.39 | 116.53 | 120.70 | 135.07 |
| | | HiRED | **39.04** | 114.09 | 136.31 | 134.85 | **142.37** | 135.14 | 134.79 | 135.07 |
| DocVQA (Metric: ANLS) | LLaVA-Mistral-7B | VL-Cache | **43** | 59 | 62 | 64 | 67 | 67 | 67 | 68 |
| | | H2O | 37 | 40 | 56 | 59 | 66 | 66 | 67 | 68 |
| | | PyramidKV | 42 | 58 | 60 | 61 | 66 | 64 | **68** | 68 |
| | | StreamingLLM | 20 | 32 | 47 | 45 | 52 | 60 | 60 | 68 |
| | | ZipCache | 41 | 40 | 60 | 45 | 65 | **68** | 67 | 68 |
| | | FastV | 18 | 26 | 38 | 49 | 63 | 64 | 67 | 68 |
| | | HiRED | 18 | 38 | 51 | 53 | 63 | 63 | 65 | 68 |
| | LLaVA-1.6-34B | VL-Cache | 40 | **82** | **84** | **85** | **85** | 85 | 85 | 85 |
| | | H2O | 28 | 41 | 75 | 82 | 82 | 84 | **85** | 85 |
| | | PyramidKV | **43** | 79 | 83 | **85** | **85** | **86** | **85** | 85 |
| | | StreamingLLM | 0 | 16 | 34 | 48 | 76 | 80 | 82 | 85 |
| | | ZipCache | 0 | 55 | 74 | 81 | **85** | 85 | **85** | 85 |
| | | FastV | 2 | 1 | 3 | 9 | 28 | 42 | 47 | 85 |
| | | HiRED | 25 | 44 | 52 | 71 | 76 | 81 | 84 | 85 |
| MathVista (Metric: Acc) | LLaVA-Mistral-7B | VL-Cache | **38** | 36 | **39** | 40 | 40 | 42 | **42** | 41 |
| | | H2O | 32 | 35 | 36 | 38 | 38 | 40 | 39 | 41 |
| | | PyramidKV | 37 | **36** | 38 | **40** | 40 | 41 | 40 | 41 |
| | | StreamingLLM | 30 | 35 | 33 | 34 | **40** | 39 | 38 | 41 |
| | | ZipCache | 35 | 35 | **39** | 34 | 40 | **42** | 40 | 41 |
| | | FastV | 25 | 29 | 37 | 38 | 39 | 40 | 41 | 41 |
| | | HiRED | 32 | 35 | 35 | 36 | **40** | 40 | 40 | 41 |
| | LLaVA-1.6-34B | VL-Cache | **41** | **42** | 42 | 44 | 45 | **45** | 42 | 43 |
| | | H2O | 37 | 40 | 40 | 39 | 42 | 44 | **44** | 43 |
| | | PyramidKV | 39 | 41 | **42** | 44 | **46** | 41 | **44** | 43 |
| | | StreamingLLM | 29 | 35 | 34 | 28 | 38 | 42 | 42 | 43 |
| | | ZipCache | 28 | 35 | 41 | 41 | 42 | 43 | 43 | 43 |
| | | FastV | 27 | 27 | 32 | 37 | 37 | 35 | 37 | 43 |
| | | HiRED | 32 | 41 | 42 | 42 | 39 | 39 | 42 | 43 |
| ChartQA (Metric: Acc) | LLaVA-Mistral-7B | VL-Cache | 20 | **33** | **41** | 38 | 40 | 40 | 41 | 41 |
| | | H2O | 23 | 26 | 32 | 39 | 41 | 40 | 41 | 41 |
| | | PyramidKV | **27** | 29 | 39 | 39 | 38 | 40 | 41 | 41 |
| | | StreamingLLM | 11 | 17 | 17 | 30 | 39 | 37 | 40 | 41 |
| | | ZipCache | 23 | **33** | 40 | **42** | 40 | 40 | 41 | 41 |
| | | FastV | 13 | 17 | 24 | 33 | 41 | 42 | **42** | 41 |
| | | HiRED | 13 | 21 | 33 | 37 | **47** | **44** | 41 | 41 |
| | LLaVA-1.6-34B | VL-Cache | 9 | 51 | **56** | 55 | **54** | 55 | **55** | 54 |
| | | H2O | 28 | 42 | 50 | 54 | **54** | 53 | 54 | 54 |
| | | PyramidKV | **31** | **52** | 54 | **56** | **54** | 54 | **55** | 54 |
| | | StreamingLLM | 5 | 7 | 11 | 14 | 43 | 51 | 52 | 54 |
| | | ZipCache | 0 | 43 | 55 | 52 | **54** | 54 | 54 | 54 |
| | | FastV | 6 | 8 | 10 | 12 | 25 | 41 | 40 | 54 |
| | | HiRED | 11 | 20 | 27 | 44 | 49 | **56** | 54 | 54 |
| TextVQA (Metric: Acc) | LLaVA-Mistral-7B | VL-Cache | 34.7 | **59.1** | 64.4 | **65.4** | 65.4 | 65.4 | 65.4 | 65.4 |
| | | H2O | **43.0** | 56.5 | 63.4 | 62.4 | **65.4** | **65.4** | **65.4** | 65.4 |
| | | PyramidKV | 35.0 | 50.1 | 60.1 | 62.8 | **65.4** | 65.4 | 65.4 | 65.4 |
| | | StreamingLLM | 19.5 | 31.4 | 43.2 | 53.6 | 59.8 | 59.9 | 61.8 | 65.4 |
| | | ZipCache | 35.6 | 51.2 | 59.4 | 63.4 | **65.4** | 65.0 | **65.4** | 65.4 |
| | | FastV | 11.1 | 38.5 | 48.2 | 61.1 | 63.4 | 61.7 | 62.7 | 65.4 |
| | | HiRED | 23.6 | 54.6 | 55.4 | 64.8 | 64.0 | 62.4 | 63.7 | 65.4 |
| | LLaVA-1.6-34B | VL-Cache | 12.9 | **70.7** | 71.4 | **73.4** | 72.4 | 73.6 | **72.7** | 72.7 |
| | | H2O | 18.3 | 47.0 | 63.2 | 69.5 | 71.5 | **74.4** | **72.7** | 72.7 |
| | | PyramidKV | **30.1** | 66.4 | 70.6 | 72.5 | 73.4 | 74.3 | **72.7** | 72.7 |
| | | StreamingLLM | 0.0 | 9.6 | 19.5 | 34.7 | 70.4 | 70.4 | 71.4 | 72.7 |
| | | ZipCache | 0.0 | 47.1 | 64.3 | 70.5 | **73.5** | 73.7 | **72.7** | 72.7 |
| | | FastV | 5.6 | 6.3 | 11.9 | 18.7 | 42.6 | 60.2 | 64.4 | 72.7 |
| | | HiRED | 27.1 | 62.1 | 66.3 | 72.9 | 69.9 | 71.2 | 71.7 | 72.7 |

Table 3: Performance of VLMs with different compression methods and datasets. With 10% KV cache, VL-Cache approaches 100% of full cache performance on most datasets and models and significantly outperformed other methods.

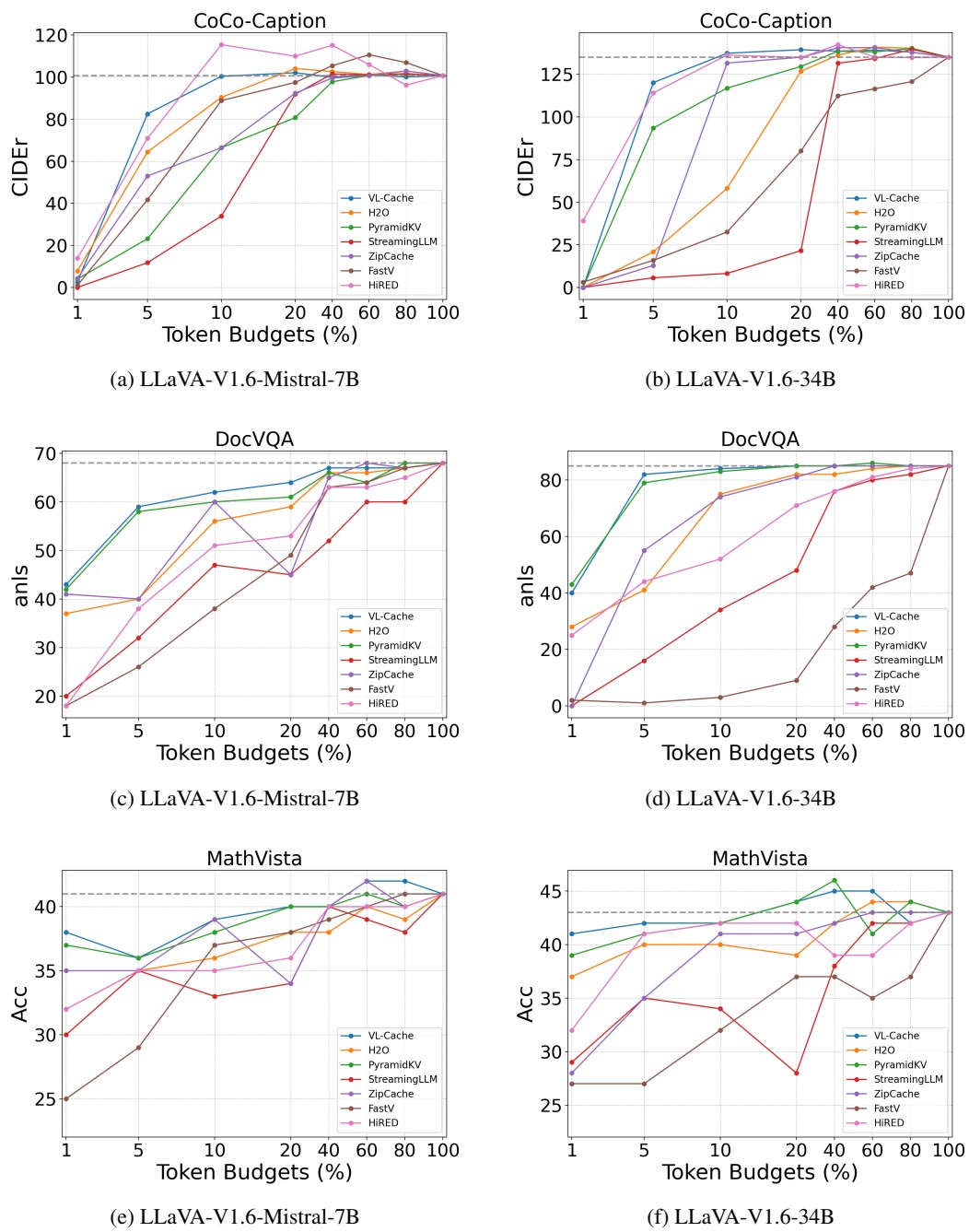

Figure 9: Evaluation results on different datasets with varied cache budgets. VL-Cache achieves comparable accuracy against full-cache and outperforms multiple baselines with limited KV cache budget

## A.4 EXTENDED RELATED WORKS

The KV cache, while essential for transformer-based LLM and VLM family, demands significant computational resources and memory, limiting inference speed. To address these challenges, researchers have explored various KV cache compression techniques, such as KV cache sparsification, quantization, or a combination of both.

**KV cache sparsification**. Heavy-Hitters (H2O) (Zhang et al., 2024b) employs cumulative attention scores to greedily evict unimportant tokens. However, this method tends to accumulate more attention on the initial tokens, introducing bias and negatively impacting the identification of key tokens

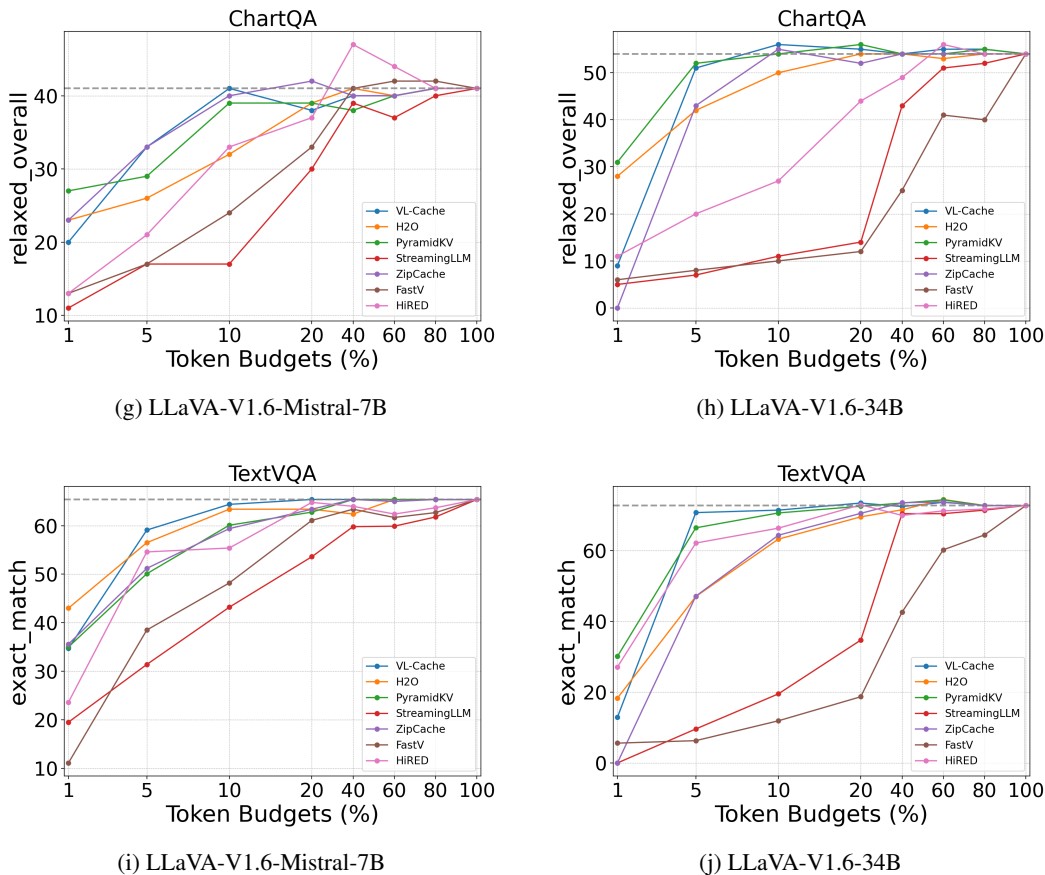

Figure 9: Additional evaluation results on different datasets with varied cache budgets, demonstrating VL-Cache's consistent performance on a wide range of settings.

during decoding. ZipCache (He et al., 2024) further normalizes the cumulative attention scores, leading to more precise prediction. Keyformer (Adnan et al., 2024) proposes a novel score function to predict the importance of each token and only window tokens and dynamic key tokens are kept in the KV cache. PyramidKV (Yang et al., 2024a) allocates progressively smaller KV cache sizes as layers get deeper. This approach can achieve an absolute accuracy improvement of nearly 20.5 on specific tasks.

**KV cache quantization**. This method reduces the size of the KV cache by utilizing lower-bit representations of the key and value pairs. In particular, FP8 and INT8 quantization of KV caches are commonly used for LLM inference. Further compression of KV cache into lower-bits is also being explored, e.g. KIVI (Liu et al., 2024b) compresses KV cache into 2-bit representations.

**Combination of quantization and sparsification**. Quantization and sparsification (Yang et al., 2024b) technically could have orthogonal implementations that can be enabled together to obtain further. However, since both are lossy methods that can impact the accuracy of downstream tasks, careful accuracy evaluation of the quality of the output need to be conducted.

**Visual Token Reduction**. In the domain of VLMs, most works focus on pruning embedded image features either before they enter the LLMs or within the hidden states of the LLMs. HiRED (Arif et al., 2024) leverages sparse attention patterns in vision transformers (e.g., CLIP-ViT) to discard visual tokens, effectively extracting key objects from images and reducing computational resources required during subsequent LLMs generation. However, it disregards textual prompts, potentially dropping essential tokens unexpectedly. FastV (Chen et al., 2025), on the other hand, prunes hidden states at a specific layer within LLMs, significantly accelerating computation in later layers. How-

ever, it applies a uniform token budget across all subsequent layers, resulting in suboptimal accuracy when the token budget is highly constrained.

The KV cache compression methods mentioned above primarily target LLMs, while in VLMs, most approaches focus on reducing visual tokens in the input image or the hidden states. And there has been limited exploration of KV cache sparsification specifically for VLMs. VL-Cache proposed in this paper analyzes and utilizes the unique sparsity pattern in VLMs, which results in better accuracy for VLM inference.

## A.5 EFFICIENT IMPLEMENTATION

In order to discard tokens that received low attention during prefill, we need to calculate 2 statistics from the attention score matrix during the prefill stage: (1) the average attention score for each token in key dimension; and (2) the count of attention scores that are smaller than $p\%$ of the maximum within the query $Q$ dimension. The **ThresholdFilter** we adopted in our algorithm is asymptotically faster than alternative methods of truncating a distribution (such as top-p and top-k), since it does not require the attention scores to be sorted.

Regular attention kernels, such as FlashAttention (Dao, 2023) and PagedAttention (Kwon et al., 2023), do not materialize the attention scores in HBM, so we need a new memory-efficient kernel to calculate the attention statistics. Overall, we aim to schedule these operations to ensure that: (1) The attention mask is not written to HBM; (2) the $QK$ product is not written to HBM; (3) the attention scores are not written to HBM; (4) similarly-parallelized operations (e.g. div and add) are fused to avoid intermediate tensors in HBM.

In the initial attempt, we used TorchCompiler with the Triton backend, which addressed requirements (1), (3), and (4), but not (2). An efficient fusion of the matmul operation with softmax is described in the FlashAttention paper and can be implemented in Triton, but TorchCompiler was not able to apply this optimization. We will now describe the optimal algorithm that satisfies all performance requirements.

The key to efficient fusion of matmul and softmax is to partition the work along the $Q$ dimension and then hold a single tile of $Q$ tensor in SRAM, while continuously loading tiles of the other tensor and computing the reductions without writing the $QK$ product back to HBM. The 2 reductions required by softmax (pre-exponentiation max, and post-exponentiation sum) are both along the $K$ dimension, so partitioning along the $Q$ dimension is required for these operations. The online softmax algorithm can be used to update the partially computed sum.

For average attention score in the KV context, we need to reduce the attention scores over the $Q$ dimension, which is not aligned with the previous computation. Instead, we need a second kernel which partitions the work over the $K$ dimension, so that a compute block can reduce over the $Q$ dimension. However, since we do not want to materialize the attention scores, the second kernel needs to recompute softmax again. In order do that, we re-use the max and sum over Q dimension from the first kernel, and these 2 tensors (both $O(Q)$ space) are the only intermediate tensors that are stored in HBM.

For sparsity ratio, counting the scores below the threshold is not possible until the maximum along the $K$ dimension is known. We can use the intermediate maximum tensor that was produced by the first kernel to count the sparse scores in the second kernel. Since each thread block will output its own count, we need the 3rd kernel to sum up the partial counts.

In summary, the optimized computation consists of 3 kernels executed serially:

1. Softmax + row-wise stats (reductions over K dimension). Reads: $Q$ and $K$ from HBM. Computes and writes to HBM:
    (a) pre-exp max over $K$ dimension (used in softmax for numerical stability)
    (b) post-exp sum over $K$ dimension (used in softmax as the normalization factor)
2. Softmax + column-wise stats (reductions over Q dimension). Reads: Q, K, pre-exp max, and post-exp sum from HBM. Computes and writes to HBM:
    (a) the post-softmax sum over the Q dimension
    (b) count of attention scores below threshold over the Q dimension

    3. Sum reduction of the partial count of attention scores below threshold

We implemented these 2 kernels in Triton and observed that each had roughly the same latency as the FlashAttention kernel, implying that this partioning is optimal. However, even with this approach, we would want to use only a few dozen tokens for querying statistics during prefill.

## A.6 SPEED BENCHMARK RESULTS

In this section, we present detailed metrics from the speed benchmark results, including exact times for both the prefill and decoding stages. We also provide an overview of the benchmarking methodology. Using Torch Profiler, we measured GPU kernel latencies and aggregated them to compute the total latency for each operation.

| Batch Size | Prompt Length | Prefill Latency (ms) | | Decoding Latency (ms) | | Speedup | | |
|---|---|---|---|---|---|---|---|---|
| | | Full Cache (Baseline) | VL-Cache (10%, Ours) | Full Cache (Baseline) | VL-Cache (10%, Ours) | Prefill Speedup | Decoding Speedup | End-to-End Speedup |
| 1 | 2k | 330.7 | 345.8 | 3821.8 | 3222.3 | 0.96 | 1.19 | 1.16 |
| 1 | 8k | 1396.1 | 1434.9 | 5877.1 | 3459.1 | 0.97 | 1.70 | 1.49 |
| 1 | 32k | 7257.2 | 7354.9 | 14247.2 | 4294.0 | 0.99 | 3.32 | 1.85 |
| 1 | 128k | 59740.9 | 60063.9 | 52547.0 | 7423.0 | 0.99 | **7.08** | 1.66 |
| 4 | 2k | 1287.0 | 1315.4 | 6156.0 | 3661.0 | 0.98 | 1.68 | 1.50 |
| 4 | 8k | 5486.5 | 5604.0 | 14229.1 | 4499.6 | 0.98 | 3.16 | 1.95 |
| 4 | 32k | 28908.7 | 29206.7 | 47321.1 | 7795.3 | 0.99 | 6.07 | 2.06 |
| 16 | 2k | 5038.3 | 5144.3 | 15078.8 | 4978.7 | 0.98 | 3.03 | 1.99 |
| 16 | 8k | 21843.6 | 22168.4 | 47688.7 | 8494.8 | 0.99 | 5.61 | 2.27 |
| 64 | 2k | 20113.2 | 20448.2 | 49666.6 | 9499.4 | 0.98 | 5.23 | **2.33** |

Table 4: Detailed performance metrics by batch size and prompt length for 100 output tokens

This breakdown of latency by GPU kernels enabled us to 1) avoid measuring CPU latency that does not relate to our method, 2) group the latency into different buckets for reporting. Empty cells mean that the inference server crashed due to insufficient GPU memory during prefill.

The following table presents the cumulative latency of all GPU kernels that are related to computing attention statistics from 50 query tokens, eviction scores for all prompt tokens, and copying the KV cache tensor after eviction of 90% of the KV cache into contiguous memory. This prefill overhead is stated in milliseconds:

| batch size | 1000 | 2000 | 4000 | 8000 | 16000 | 32000 | 64000 | 128000 |
|---|---|---|---|---|---|---|---|---|
| 1 | 5.20 | 7.20 | 10.10 | 17.30 | 28.30 | 49.70 | 100.20 | 189.60 |
| 2 | 6.40 | 9.50 | 15.80 | 29.00 | 51.80 | 92.50 | 188.50 | |
| 4 | 8.50 | 15.20 | 27.40 | 52.50 | 93.30 | 175.80 | | |
| 8 | 13.70 | 26.80 | 50.90 | 96.60 | 178.40 | | | |
| 16 | 25.20 | 51.30 | 95.60 | 183.50 | | | | |
| 32 | 48.50 | 97.10 | 182.60 | | | | | |
| 64 | 91.60 | 186.70 | | | | | | |

We define the prefill latency as the latency of generating 1 output token – starting with input token IDs and ending with output logits, not including tokenization and sampling. The following table presents the prefill speedup ($\frac{\text{prefill}+\text{overhead}}{\text{prefill}}$). We observe that the overhead is never more than 6% and becomes smaller as the prompt gets longer:

| batch size | 1000 | 2000 | 4000 | 8000 | 16000 | 32000 | 64000 | 128000 |
|---|---|---|---|---|---|---|---|---|
| 1 | 0.94 | 0.96 | 0.97 | 0.97 | 0.99 | 0.99 | 0.99 | 0.99 |
| 2 | 0.96 | 0.97 | 0.97 | 0.98 | 0.98 | 0.99 | 0.99 | |
| 4 | 0.97 | 0.98 | 0.98 | 0.98 | 0.98 | 0.99 | | |
| 8 | 0.98 | 0.98 | 0.98 | 0.99 | 0.99 | | | |
| 16 | 0.98 | 0.98 | 0.99 | 0.99 | | | | |
| 32 | 0.98 | 0.98 | 0.98 | | | | | |
| 64 | 0.99 | 0.98 | | | | | | |

We calculated decoding latency as the difference between end-to-end latency and the prefill latency. Since we use static batching with a constant number of output tokens, it's easy to decompose the end-to-end latency into its components without concerns for queue waiting, scheduler stalling, sequence termination, etc. The following table presents the decoding speedup ($\frac{\text{decoding latency with full KV cache}}{\text{decoding latency with 10\% of KV cache}}$):

| batch size | 1000 | 2000 | 4000 | 8000 | 16000 | 32000 | 64000 | 128000 |
|---|---|---|---|---|---|---|---|---|
| 1 | 1.10 | 1.19 | 1.37 | 1.70 | 2.31 | 3.32 | 5.17 | 7.08 |
| 2 | 1.18 | 1.36 | 1.68 | 2.27 | 3.16 | 4.56 | 6.78 | |
| 4 | 1.36 | 1.68 | 2.26 | 3.16 | 4.39 | 6.07 | | |
| 8 | 1.73 | 2.29 | 3.13 | 4.37 | 5.87 | | | |
| 16 | 2.19 | 3.03 | 4.18 | 5.61 | | | | |
| 32 | 2.94 | 4.12 | 5.53 | | | | | |
| 64 | 3.83 | 5.23 | | | | | | |

The end-to-end speedup is calculated as $\frac{\text{Baseline latency}}{\text{VLCache latency}}$. VLCache latency includes prefill latency, VLCache overhead, and 99 decoding passes to generate a total of 100 output tokens:

| batch size | 1000 | 2000 | 4000 | 8000 | 16000 | 32000 | 64000 | 128000 |
|---|---|---|---|---|---|---|---|---|
| 1 | 1.09 | 1.16 | 1.30 | 1.49 | 1.71 | 1.85 | 1.89 | 1.66 |
| 2 | 1.16 | 1.30 | 1.49 | 1.73 | 1.89 | 1.97 | 1.94 | |
| 4 | 1.30 | 1.50 | 1.74 | 1.95 | 2.06 | 2.06 | | |
| 8 | 1.54 | 1.79 | 2.00 | 2.17 | 2.23 | | | |
| 16 | 1.75 | 1.99 | 2.17 | 2.27 | | | | |
| 32 | 1.98 | 2.19 | 2.32 | | | | | |
| 64 | 2.18 | 2.33 | | | | | | |

## A.7 MEASURING ATTENTION TO VISUAL AND LANGUAGE TOKENS

As we have established in the Section 3.1, the attention sparsity during decoding can be predicted from attention scores in prefill phase. To further understand the importance of visual tokens and language tokens in the prompt, we examine the division of attention from decoding tokens to these two modalities of prompt tokens.

We propose **Contribution** for quantitative analysis and visualization of the *layer-wise modality-specific* attention patterns. Let $T$ be the current sequence length, $t$ be the index of the first decoded token, and $A^{(l)} \in \mathbb{R}^{T \times T}$ be the attention matrix at layer $l$ for one specific head.

$$\textbf{Contribution}_{mod}^{(l)} := \frac{1}{T-t+1} \sum_{i \geq t} \frac{\sum_{j \in J_{mod}} \textbf{ThresholdFilter}(A^{(l)}, p)_{ij}}{\sum_{j \in J_{all}} \textbf{ThresholdFilter}(A^{(l)}, p)_{ij}} \tag{3}$$

Here, we use $J_{mod}$ and $J_{all}$ to denote KV cache indices of one selected modality (language or vision) and all modalities respectively in the input prompt. As shown in Figure 10, VLMs allocate primary attention to visual tokens in the first layer, while in the second layer, the contributions of visual and language tokens are comparable. From the third layer onward, language tokens dominate, with a slight increase in visual token contribution in the middle layers. Additionally, we also propose **Coverage** to analyze the ratio of the number of tokens from a specific modality, defined as follows:

$$\textbf{Coverage}_{mod}^{(l)} := \frac{1}{T-t+1} \sum_{i \geq t} \frac{\sum_{j \in J_{mod}} \textbf{TopK}(A^{(l)}, k)_{ij}}{\sum_{j \in J_{all}} \textbf{TopK}(A^{(l)}, k)_{ij}} \tag{4}$$

Specifically,

$$\textbf{TopK}(A, k)_{ij} = \mathbb{1}[A_{ij} \in \{A_i^{(r-k+1)}, A_i^{(r-k+2)}, ..., A_i^{(r)}\}] \tag{5}$$

Here, we define $k$ as $\lfloor \alpha \cdot T \rfloor$, $r$ as the number of columns of matrix $A$, and overload notations to use $A_i^{(n)}$ as the $n$-th order statistic of the row vector. Additionally, $\alpha \in (0, 1)$ denotes the token budget threshold (e.g. $\alpha = 10\%$ means only the 10% of sequence length of KV cache is retained). After KV cache compression with **TopK** selection, we assess the ratio of visual attention to language attention among the remaining tokens. In Figure 11, with $\alpha = 10\%$ applied, a similar trend in coverage is observed compared to Figure 10. Notably, in the middle layer, visual tokens constitute a larger proportion than their contribution, indicating that the contribution per token is smaller.

Furthermore, since contribution of language and visual tokens differ by layer, the optimal budget for each layer may depend on the ratio of visual to language tokens in the prompt. In contrast, previous pyramid-style cache allocation methods do not adapt to the input prompts at inference time.

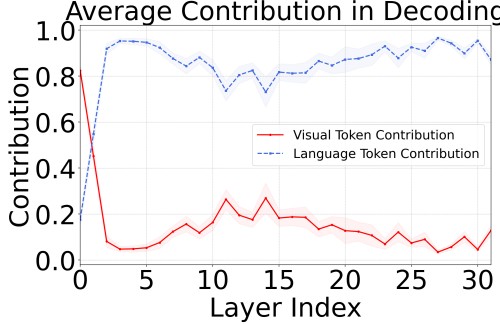

Figure 10: Contribution

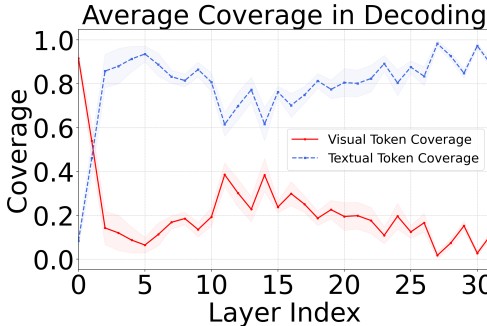

Figure 11: Coverage

