# OpenReview forum: "VL-Cache: Sparsity and Modality-Aware KV Cache Compression for Vision-Language Model Inference Acceleration"
_ICLR.cc/2025/Conference — ICLR 2025 Poster_

### Official Review · Reviewer_X8F2 · 2024-10-28

**Soundness:** 3
**Presentation:** 3
**Contribution:** 2
**Rating:** 6
**Confidence:** 4

**Summary:**

The paper tackles the challenge of accelerating vision-language models (VLMs) by leveraging a sparse Key-Value (KV) cache. Unlike existing compression methods for large language models (LLMs), which are less effective for VLMs, the proposed VL-Cache takes advantage of unique sparsity patterns in visual and text tokens. It uses layer-adaptive budget allocation to allocate cache resources based on sparsity levels and modality-aware token scoring to determine token importance with post-vision text tokens. By retaining only the most crucial cache entries, VL-Cache effectively reduces memory usage and latency. Experimental results on three lmms-eval tasks demonstrate the method’s effectiveness.

**Strengths:**

The paper is well-written and easy to follow. The sparsity analysis of prefilling and decoding effectively motivates the use of adaptive sparsity. The proposed method achieves significant speedup in decoding, particularly with long prefill tokens.

**Weaknesses:**

Please see the questions.

**Questions:**

1.	Some notations in the proposed method lack clarity. For instance, the meanings of $\Gamma$ and $Z$ are not sufficiently explained, making it difficult to understand their roles in the algorithm. Providing more detailed explanations would improve the clarity of the presentation.

2.	The authors state that the maximum batch size is constrained by peak memory usage during prefill rather than by the KV cache size, suggesting that compressing the KV cache does not increase the batch size. Does this imply that the proposed KV cache compression does not actually reduce overall memory usage? If so, why is that the case? This raises concerns about whether the title “KV Cache Compression” accurately reflects the method’s ability to save GPU memory, as the proposed approach appears to be more of a sparse computation technique applied during decoding.

3.	In Table 3, the reported speedup is based on generating a total of 100 output tokens. It would be helpful if the authors could provide the exact prefill and decoding times separately. Given that with a prompt length of 128k and only 100 output tokens, the prefill time is likely to dominate the overall processing time, having this detailed breakdown would clarify the impact of the proposed method on different stages of the inference process.

4.	In Section 4.2, the authors point out that accumulated attention has the limitation of assigning disproportionately high scores to earlier tokens due to summation over the full query dimension. To address this, they propose accumulated post-vision attention. However, there is another approach [1] that addresses the same issue using normalized attention. It would be helpful if the authors could include more discussion on this alternative method and provide additional experimental comparisons to better illustrate the benefits of their approach.

5.	The experimental evaluation is insufficient, as it lacks comparisons with two important baselines for VLM acceleration: FastV [1] and HiRED [2]. Additional comparisons with these methods are needed to strengthen the evaluation.

6.	Although the proposed method targets vision-language models, the current evaluation lacks coverage of a comprehensive range of tasks. To better demonstrate its effectiveness, it would be beneficial for the authors to include additional results on video benchmarks, such as Video-MME [4]. This would help showcase the method’s applicability to video tasks.

Reference:

[1] ZipCache: Accurate and Efficient KV Cache Quantization with Salient Token Identification. NeurIPS 2024.

[2] An image is worth 1/2 tokens after layer 2: Plug-and-play inference acceleration for large vision-language models. ECCV 2024.

[3] HiRED: Attention-Guided Token Dropping for Efficient Inference of High-Resolution Vision-Language Models in Resource-Constrained Environments. arXiv 2024.

[4] Video-MME: The First-Ever Comprehensive Evaluation Benchmark of Multi-modal LLMs in Video Analysis. arXiv 2024.

---

> ### Author Response · Authors · 2024-11-24
> **Response to Reviewer X8F2**
>
> We thank the reviewer for carefully reviewing our paper, recognizing our paper as well-written and easy-to-follow, our method is well-motivated by sparsity analysis and achieves significant decoding speedups. We also appreciate the insightful questions and constructive feedbacks from the reviewer and have been working diligently over the last week. We address them as follows.
>
>
> ### On Question 1
>
> We apologize for the confusion from our notations. Specifically, $\Gamma$ is a 2D array storing the sparsity at each head and layer, while $Z$ is the sum of non-sparse ratios across layers as the normalizer when computing cache budget. We have added in-line annotations to Algorithm 1 and references from the main text to the algorithm to improve the clarity of our presentation.
>
>
> ### On Question 2
>
> These are great questions. Our method does reduce the overall memory usage as we can evict 90% of KV cache tokens while maintaining great performance. Our method does not reduce peak memory usage during prefill because we propose to evict tokens after prefill is finished so that we can allocate an appropriate cache budget based on each request, whose effectiveness is demonstrated by the consistently strong performance across models and tasks. While our method can not use a higher static batch size at the moment, it can effectively increase the batch size when continuous batching, a widely used technique in LLM serving, is utilized. Intuitively, what continuous batching does is to start processing a new request whenever a request in the current batch finishes. With KV cache compression, the GPU memory usage is reduced significantly once decoding starts. Thus, while the model is decoding current requests with much fewer KV cache, new requests can be processed, which results in higher batch size and boosted throughput. Given these reasons, we think “KV Cache Compression” still well characterize our work.
>
>
> ### On Question 3
>
> The reviewer brought up an excellent point. We are currently waiting to perform the benchmark again due to resource constraints, and plan to provide the exact prefill and decoding times by Monday. We appreciate the understanding of the reviewer. Meanwhile, we would like to first point the reviewer to the End-to-End Speedup column of Table 2, where we showed that significant decoding speedups lead to end-to-end (prefill + decoding) speedups across prompt lengths.
>
>
> ### On Question 4
>
> We thank the reviewer for bringing this highly relevant work and have provided discussions and results in the overall reply to all reviewers.

---

> ### Author Response · Authors · 2024-11-24
> **Response to Reviewer X8F2**
>
> ### On Question 5
>
> We thank the reviewer for pointing us to these two interesting works that we previously overlooked, both of which focus on evicting visual tokens based on vision encoder’s attention patterns before they are used by the LLM. We show comparison with both methods below and included them in Appendix A.2 of the revised paper as well. While HiRED outperforms our method on CoCo-Caption for LLaVA-Mistral-7B, it underperforms on all other cases, especially on DocVQA. This large performance fluctuation implies that HiRED’s strategy of dropping visual tokens before LLM without considering the language query could lead to loss of important information. Our method consistently outperforms FastV since it only focuses on hidden states pruning and ignores the huge memory consumption of KV cache. As a result, under the same memory constraint, our method naturally retains accuracy better.
>
> Dataset | Model | Method | 1% | 5% | 10% | 20% | 40% | 60% | 80% | 100%
> --- | --- | --- | --- | --- | --- | --- | --- | --- | --- | ---
> Coco-Caption (Metric: CIDEr) | LLaVA-Mistral-7B | VL-Cache | 2.64 | **82.53** | 100.36 | 102.06 | 99.93 | 101.07 | 100.08 | 100.68
> Coco-Caption (Metric: CIDEr) | LLaVA-Mistral-7B | FastV | 9.98 | 41.17 | 88.82 | 97.47 | 105.40 | 110.60 | **106.90** | 100.68
> Coco-Caption (Metric: CIDEr) | LLaVA-Mistral-7B | HiRED | **13.87** | 70.90 | **115.37** | **109.91** | **115.09** | **115.09** | 105.93 | 100.68
> Coco-Caption (Metric: CIDEr) | LLaVA-1.6-34B | VL-Cache | 0.00 | **120.1**1 | **137.35** | **139.42** | 138.58 | **139.19** | **138.01** | 135.07
> Coco-Caption (Metric: CIDEr) | LLaVA-1.6-34B | FastV | 3.10 | 15.86 | 32.55 | 80.04 | 112.39 | 116.53 | 120.70 | 135.07
> Coco-Caption (Metric: CIDEr) | LLaVA-1.6-34B | HiRED | **39.04** | 114.09 | 136.31 | 134.85 | **142.37** | 135.14 | 134.79 | 135.07
> DocVQA (Metric: ANLS) | LLaVA-Mistral-7B | VL-Cache | **43** | **59** | **62** | **64** | **67** | **67** | **67** | 68
> DocVQA (Metric: ANLS) | LLaVA-Mistral-7B | FastV | 18 | 26 | 38 | 49 | 63 | 64 | **67** | 68
> DocVQA (Metric: ANLS) | LLaVA-Mistral-7B | HiRED | 18 | 38 | 51 | 53 | 63 | 63 | 65 | 68
> DocVQA (Metric: ANLS) | LLaVA-1.6-34B | VL-Cache | **41** | **82** | **84** | **85** | **85** | **85** | **85** | 85
> DocVQA (Metric: ANLS)| LLaVA-1.6-34B | FastV | 2 | 1 | 3 | 9 | 28 | 42 | 47 | 85
> DocVQA (Metric: ANLS)| LLaVA-1.6-34B | HiRED | 25 | 44 | 52 | 71 | 76 | 81 | 84 | 85
> MathVista (Metric: ACC) | LLaVA-Mistral-7B | VL-Cache | **38** | **36** | **39** | **40** | **40** | **42** | **42** | 41
> MathVista (Metric: ACC) | LLaVA-Mistral-7B | FastV | 25 | 29 | 37 | 38 | 39 | 40 | 41 | 41
> MathVista (Metric: ACC) | LLaVA-Mistral-7B | HiRED | 32 | 35 | 35 | 36 | **40** | 40 | 40 | 41
> MathVista (Metric: ACC) | LLaVA-1.6-34B | VL-Cache | **41** | **42** | **42** | **44** | **45** | **45** | **42** | 43
> MathVista (Metric: ACC) | LLaVA-1.6-34B | FastV | 27 | 27 | 32 | 37 | 37 | 35 | 37 | 43
> MathVista (Metric: ACC) | LLaVA-1.6-34B | HiRED | 32 | 41 | **42** | 42 | 39 | 39 | 42 | 43
>
>
> ### On Question 6
>
> We agree that video benchmarks would be a great addition to showcase the effectiveness of our method. However, the evaluation on Video-MME is taking extended time and we will provide an update by Monday. We appreciate the patience and understanding of the reviewer as we try our best to prepare the result.

---

> ### Comment · Reviewer_X8F2 · 2024-11-25
> **Official Comment by Reviewer X8F2**
>
> Thank you to the authors for their response. I look forward to the additional results and the feedback from the other reviewers.

---

> > ### Author Response · Authors · 2024-11-26
> > **Response to Reviewer X8F2**
> >
> > Dear Reviewer X8F2,
> >
> > Thank you for your early engagement and your patience! We would like to follow up on the remaining results that were requested.
> >
> > ### On Question 3
> >
> > Here, we report the exact prefill and decoding time as requested. We have listed the detailed values below and included them to Table 4 in Appendix A.6.
> >
> > | batch_size | context_len | attention_impl | prefill_latency_ms | decoding_latency_ms | total_latency_ms |
> > |-------------:|--------------:|:---------------------|---------------------:|----------------------:|-------------------:|
> > | 1 | 2k | Full KV Cache | 330.7 | 3821.8 | 4152.5 |
> > | 1 | 2k | VLCache (10% budget) | 345.8 | 3222.3 | 3568.1 |
> > | 1 | 8k | Full KV Cache | 1396.1 | 5877.1 | 7273.2 |
> > | 1 | 8k | VLCache (10% budget) | 1434.9 | 3459.1 | 4894 |
> > | 1 | 32k | Full KV Cache | 7257.2 | 14247.2 | 21504.4 |
> > | 1 | 32k | VLCache (10% budget) | 7354.9 | 4294 | 11648.9 |
> > | 1 | 128k | Full KV Cache | 59740.9 | 52547 | 112288 |
> > | 1 | 128k | VLCache (10% budget) | 60063.9 | 7423 | 67486.9 |
> > | 4 | 2k | Full KV Cache | 1287 | 6156 | 7443 |
> > | 4 | 2k | VLCache (10% budget) | 1315.4 | 3661 | 4976.4 |
> > | 4 | 8k | Full KV Cache | 5486.5 | 14229.1 | 19715.6 |
> > | 4 | 8k | VLCache (10% budget) | 5604 | 4499.6 | 10103.6 |
> > | 4 | 32k | Full KV Cache | 28908.7 | 47321.1 | 76229.8 |
> > | 4 | 32k | VLCache (10% budget) | 29206.7 | 7795.3 | 37002 |
> > | 16 | 2k | Full KV Cache | 5038.3 | 15078.8 | 20117.1 |
> > | 16 | 2k | VLCache (10% budget) | 5144.3 | 4978.7 | 10123 |
> > | 16 | 8k | Full KV Cache | 21843.6 | 47688.7 | 69532.3 |
> > | 16 | 8k | VLCache (10% budget) | 22168.4 | 8494.8 | 30663.2 |
> > | 64 | 2k | Full KV Cache | 20113.2 | 49666.6 | 69779.8 |
> > | 64 | 2k | VLCache (10% budget) | 20448.2 | 9499.4 | 29947.6 |

---

> > > ### Author Response · Authors · 2024-11-26
> > > **Response to Reviewer X8F2**
> > >
> > > ### On Question 6
> > >
> > > We show the results when evaluating our method and several other baselines on Video-MME dataset. We used LLaVA-NeXT-Video-7B-32K [1] model to conduct experiments as we found that LLaVA-1.6-Mistral-7B performs poorly on Video-MME (perception score of 9 compared to 39 of LLaVA-NeXT),  due to the fact that it was not fine-tuned on video data.
> > >
> > > Most KV cache compression methods, except for FastV, have not been tested on video tasks, but can be applied to video inputs due to generality of their formulation. However, for HiRED, we found it quite challenging to extend this method to accommodate video inputs efficiently since it operates on each image separately, so we do not report its result. We were able to reproduce FastV in our repo for LLaVA-NeXT-Video-7B-32K. But, since it requires materializing the full attention matrix for two layers, FastV suffers from a memory bottleneck and gets OOM error when using all frames. As a result, we subsampled 32 frames from each video for all methods for a fair comparison.  This issue was not surfacing in the FastV paper since the authors used LLaVA-1.5 [2] model, which uses a fixed amount of 576 visual tokens for all images no matter their resolution, and thus leads to much shorter input lengths. In contrast, our method only materializes the post vision attention matrix, which requires much fewer memory, and is compatible with FlashAttention [3] to support much higher input lengths.
> > >
> > > In the following table, we compare VL-Cache against Streaming LLM, H2O, ZipCache, and FastV. When analyzing the results, we found that all methods, except for FastV, achieve full cache performance no matter the cache budget. It turns out the reason is that Video-MME only requires one output token, the letter answer to the multiple choices question. In this case, KV cache compression methods, with their current implementation, will recover full cache performance since the first token is essentially generated using full cache. For example, H2O evict tokens after prefill hidden states from a layer is computed, while our method evicts tokens until prefill hidden states from the last layer is computed.
> > >
> > > Dataset | Model | Method | 1% | 5% | 10% | 20% | 40% | 60% | 80% | 100%
> > > --- | --- | --- | --- | --- | --- | --- | --- | --- | --- | ---
> > > Video-MME (Metric: Perception Score) | LLaVA-NeXT-Video-7B-32K | VL-Cache | 39 | 39 | 39  | 39 | 39 | 39 | 39 | 39|
> > > Video-MME (Metric: Perception Score) | LLaVA-NeXT-Video-7B-32K | H2O | 39 | 39 | 39  | 39 | 39 | 39 | 39 | 39|
> > > Video-MME (Metric: Perception Score) | LLaVA-NeXT-Video-7B-32K | PyramidKV | 39| 39 | 39  | 39 | 39 | 39 | 39 | 39|
> > > Video-MME (Metric: Perception Score) | LLaVA-NeXT-Video-7B-32K | StreamingLLM | 39| 39 | 39  | 39 | 39 | 39 | 39 | 39|
> > > Video-MME (Metric: Perception Score) | LLaVA-NeXT-Video-7B-32K | ZipCache | 39| 39 | 39  | 39 | 39 | 39 | 39 | 39|
> > > Video-MME (Metric: Perception Score) | LLaVA-NeXT-Video-7B-32K | FastV | 30 | 40 | 38  | 40 | 39 | 39 | 39 | 39|
> > >
> > > We would like to point out that while videos can be essentially viewed as multiple images, they might demonstrate very different attention patterns and present unique KV cache compression opportunities due to their temporal dependencies. We believe our work is a meaningful step to the direction of designing modality-aware KV cache compression methods for more efficient VLM inference. We plan to dive deeper to VLM for video tasks in future work and have listed it as an opportunity in the Conclusion section of our paper.
> > >
> > >
> > > [1] Liu, Haotian, et al. "Llava-next: Improved reasoning, ocr, and world knowledge." Jan. 2024.
> > >
> > > [2] Liu, Haotian, et al. "Improved baselines with visual instruction tuning." Proceedings of the IEEE/CVF Conference on Computer Vision and Pattern Recognition. 2024.
> > >
> > > [3] Dao, Tri, et al. "Flashattention: Fast and memory-efficient exact attention with io-awareness." Advances in Neural Information Processing Systems 35 (2022): 16344-16359.

---

> > > > ### Comment · Reviewer_X8F2 · 2024-11-27
> > > > **Official Comment by Reviewer X8F2**
> > > >
> > > > Thank you for providing the additional results. The authors have adequately addressed my concerns, and I will raise my score.

---

> > > > > ### Author Response · Authors · 2024-11-27
> > > > >
> > > > > Thank you for your reply Reviewer X8F2. We are glad that our response has sufficiently addressed your concerns and appreciate that you have raised the score!

---

### Official Review · Reviewer_KiPr · 2024-10-29

**Soundness:** 2
**Presentation:** 3
**Contribution:** 2
**Rating:** 6
**Confidence:** 4

**Summary:**

The authors aim to accelerate VLMs by reusing KV caches for encoding visual contexts. The authors observe the limitations of existing KV cache compression methods and propose a specific version for VLMs. Based on the observation of sparsity patterns in prefilling and decoding stages, the main idea is to design a cache allocation method that balances the cache size and model accuracy. Evaluations based on vision-language benchmarks show the performance of model accuracy and speedups.

**Strengths:**

1.	The authors identify the I/O bottleneck between GPU’s HBM and SRAM in the decoding stage of VLMs, motivating the innovative design of KV cache compression. This understanding helps to address the challenge of scaling VLMs and optimizing their performance.
2.	Through Figure 1, the authors illustrate the distinctive attention sparsity patterns between VLMs and LLMs, emphasizing the need for modality-aware compression techniques tailored for VLMs.
3.	I have learned a lot from the author's observations. The KV cache budget and compression ratios differ in layers and modalities. This provides good insights into the optimization of VLM cache compression.
4.	To compress the VLM cache, the authors propose two key methods: a sparsity-aware cache budget allocation strategy and a modality-aware token scoring policy, to optimize the cache budget allocation and token score measurement, respectively.
5.	The experimental results show that the proposed methods achieve significant inference speedups and GPU memory savings.

**Weaknesses:**

1.	In Figure 1, the authors measure attention scores to verify the modality boundary, but the specific methodology for calculating these scores remains unclear. Please provide additional details on the measurement process.
2.	The term “post-vision attention” is frequently used but not explicitly defined, making the relevant sentences difficult to follow (“the attention patterns from the output language tokens is much closer aligned with the language tokens that follow the visual tokens in the prompt (the post-vision attention) rather than the visual tokens themselves”). A clear description of post-vision attention at its first occurrence would enhance the paper's readability.
3.	The threshold filter in Section 3.1 utilizes the hyper-parameter p to control sparsification. Conducting ablation studies on the impact of different p values would provide deeper insights into the robustness and sensitivity of the proposed method.
4.	To Algorithm 1, while it dynamically allocates the cache budget based on each prompt, its computational complexity raises my concern. Please analyze the computational overhead (in lines 11 to 14).
5.	Besides, Algorithm 1 does not explicitly reflect the modality-aware design as has been highlighted in this paper. Please clarify how the algorithm incorporates modality properties.
6.	In experiments, the I/O traffic amount caused by KV caching should be reported.

**Questions:**

Please refer to the questions in the weakness part.

---

> ### Author Response · Authors · 2024-11-24
> **Response to Reviewer KiPr**
>
> We thank the reviewer for the thoughtful review and for the recognition that our method is well-motivated, provides good insights into the optimization of VLM cache compression, and achieves significant inference speedups and GPU memory savings. We would like to address the concerns and questions raised by the reviewer as follows:
>
>
> ### On Weakness 1
>
>
> The way we collect the attention score matrix is actually quite straightforward. Let $Q, K \in R^{n \times d}$ be the query and key matrices respectively, where n includes both prompt and decoded tokens, the attention score matrix $A \in R^{n \times n}$ is computed as $softmax(QK^{T} / \sqrt{d})$. We note that the decoded tokens are generated by the original VLM without any KV cache compression, so the attention pattern during decoding stage faithfully reflects the true importance of prompt tokens.
>
>
> ### On Weakness 2
>
>
> We thank the reviewer for this excellent suggestion. We have added a formal definition of post vision attention in its first occurrence to improve the readability of our paper. For reference, we formally define $\textbf{post-vision attention}$ as the sub attention score matrix that's sliced along the query dimension to only include language prompt tokens after vision tokens, as illustrated in Figure 1(b) and the updated Figure 4.
>
> ### On Weakness 3
>
> We thank the reviewer for the great suggestion. We have conducted ablation studies on the impact of the hyperparameter $p$ for Llava-Mistral-7B on the CoCo-Caption dataset, where we vary thresholds $p$ from 0.0001 to 0.1. We found that our method is robust to $p$ in general, and more so when the token budgets are higher. We have also added this analysis to Appendix A.2 of our paper.
>
> | Token Budget (%) | $p$ = 0.0001 | $p$ = 0.001 | $p$ = 0.01 | $p$ = 0.1 |
> |------------------|------------|-----------|----------|----------|
> | 10 | 101.36 | 97.78 | 98.51 | 98.77 |
> | 20 | 98.60 | 97.06 | 99.87 | 101.24 |
> | 40 | 99.45 | 99.6008 | 99.5106 | 99.12 |
> | 60 | 101.04 | 100.90 | 100.71 | 100.10 |
> | 80 | 103.02 | 101.76 | 101.08 | 101.76 |
>
>
> ### On Weakness 4
>
> The reviewer raises an insightful point that dynamically allocating cache budget based on each prompt can be computationally expensive. Fortunately, we were able to implement a highly efficient Triton-kernel that we described in detail in Appendix A.5 of our paper. We have also showed in the column “Prefill Speedup” of Table 2 in our paper that the incurred overhead is only around 1% to 4% of prefill time. We hope this information from the paper could relieve the reviewer’s concern over the computational overhead of our method.
>
>
> ### On weakness 5
>
> We agree with the reviewer that we did not emphasize enough how Algorithm 1 incorporates modality properties. We would like to clarify that the `ComputeSparsity` helper method, which we have renamed to `ComputePostVisionSparsity` in our updated manuscript, leverages the post vision attention matrix, instead of the full attention matrix, to compute the attention head sparsity. Previously, we showed that post vision attention patterns are much more similar to decoding attention patterns. Thus, using post vision attention patterns to compute sparsity also better captures the necessary cache budget required to preserve performance.
>
>
> ### On weakness 6
>
> We thank the reviewer for this suggestion. We would like to request more context regarding the insights the reviewer aims to extract with this I/O traffic information. In case this is helpful, attention kernels load KV context once, so reduction in the number of KV tokens translates to reduction in IO traffic between HBM and Shared Memory of GPU. Since typical VLM & LLM inference engine store and retrieve the KV cache for the entire prompt in HBM, our method significantly reduces KV cache I/O.

---

> > ### Comment · Reviewer_KiPr · 2024-11-27
> > **Post-rebuttal Comments**
> >
> > The authors have addressed my most concerns and I will keep my score.

---

> > > ### Author Response · Authors · 2024-11-27
> > >
> > > Thank you Reviewer KiPr for confirming our response has addressed your concerns!

---

### Official Review · Reviewer_XVzL · 2024-10-31

**Soundness:** 3
**Presentation:** 3
**Contribution:** 2
**Rating:** 6
**Confidence:** 5

**Summary:**

This paper introduces a modality-aware KV cache compression method for Vision-Language Models (VLM). It introduces a dynamic cache budget allocation mechanism and employs post-vision language tokens to compute attention scores.

**Strengths:**

The insight that the attention patterns of generated tokens closely align with language tokens is valuable.

**Weaknesses:**

1.	Algorithm 1 only outlines two procedures without detailing how these procedures are integrated into the algorithm.
2.	Definition 3.1 and the following text is unclear. The terms Psi(S) and S_Psi are used without clear explanation. A precise definition of Cache Hit Rate is essential.
3.	Figure 4 lacks clarity. It does not effectively illustrate how the budget is allocated or how post-vision attention is utilized.
4.	In the related work, ZipCache proposes a policy using normalized attention scores. This should be considered and compared.
5.	The KV cache for generated tokens is not addressed.
6.	The performance of VL-Cache is not convincingly demonstrated in Table 2. VL-Cache does not significantly outperform other methods, while H2O has achieved the strongest performance on many benchmarks. It is recommended to test VL-Cache on more challenging benchmarks.

**Questions:**

See weaknesses above.

---

> ### Author Response · Authors · 2024-11-24
> **Response to Reviewer XVzL**
>
> We thank the reviewer for their recognition of our work with the comment “the insight that the attention patterns of generated tokens closely align with language tokens is valuable”. We understand their concerns and address them as follows.
>
> ### On Weakness 1
>
> To better contextualize the procedures in Algorithm 1 to our core methodology, we made two improvements: 1. we relate the algorithm described in the main context to specific lines of Algorithm 1, and 2) we added comments in Algorithm 1 to explicitly note procedure `SkewedCacheBudgetAllocation` as the main method to compute layer-wise budget and procedure `ComputeSparsity`, which we renamed to `ComputePostVisionSparsity` in our edited version for clarity, as the helper method that utilizes post-vision attention to compute sparsity.
>
>
> ### On Weakness 2
>
> We echo the reviewer’s point that Cache Hit Rate should be clearly defined and explained. We use $\psi(S)$ to indicate the token scores assigned to $S$ with a policy $\psi$, and $S_{\psi, k}$ (changed from $S_\psi$ for clarity) to indicate the top-k indices selected by $\psi$. We have added these explanations in the main text. Additionally, we recognized that we defined two relevant concepts, $S$, the indices of cache tokens, and $\psi$, the scoring policy that maps token indices to scalar values as their importance, much earlier in Section 2.2, which might have added burdens on readers. Thus, we briefly re-state their definitions and refer the readers to Section 2.2 for further details before Definition 3.1. We hope these adjustments improve the readability of our paper.
>
>
> ### On Weakness 3
>
> We thank the reviewer for the constructive feedback. We have made major changes to Figure 4 to better illustrate our methodology and included in our updated paper. During prefill, post-vision attention matrices are utilized to compute sparsity-driven layer-wise budgets and rank the importance of cache tokens. Unimportant tokens are then evicted, allowing lower memory usage and accelerated decoding with reduced KV cache.
>
>
> ### On Weakness 4
>
>
> We thank the reviewer for bringing this highly relevant work to our attention and have provided comparisons with ZipCache’s scoring policy in our overall reply to all reviewers.
>
>
> ### On Weakness 5
>
> The reviewer has raised an insightful point that we did not compress KV cache for decoded tokens . While we plan to tackle this issue initially, an analysis on input and output lengths of current vision-language tasks reveals that the ratio prompt tokens to output tokens is quite high (median ~320x across datasets). As a result, compressing KV cache of newly generated tokens provides limited memory savings. By focusing on compressing prefill KV cache, we addressed the most pressing memory bottleneck for VLM inference tasks. Still, we anticipate the need for compressing decoding KV cache will arise when more diverse use cases emerge for VLMs. Therefore, we have listed this limitation of our current method in the Conclusion section and mentioned a periodical compression strategy for newly generated tokens to strike a balance between latency and memory saving.
>
> ### On Weakness 6
>
> We respectfully disagree with the reviewer’s claim that the VL-Cache does not significantly outperform other methods. With 10% KV cache, VL-Cache achieves 91% - 100% of full cache performance on most datasets and models and significantly outperformed other methods. In particular, under the same cache budget, H2O only achieves 43% - 93% of full cache performance. Since all KV cache compression methods will converge to full cache performance as the more cache budget is granted, we believe strong performance at low cache budgets is a clear testament of the effectiveness of VL-Cache in retaining important KV tokens. Meanwhile, we recognize the challenges of interpreting the table as we included results for various cache budgets. Therefore, we have added more information in the table’s caption to provide guidance.

---

> ### Author Response · Authors · 2024-11-24
> **Response to Reviewer XVzL**
>
> Finally, we have tested VL-Cache on two more datasets, ChartQA and TextVQA. Again, we found that with 10% KV cache, VL-Cache manages to approach full-cache performance on both datasets while outperforming all other baselines.
>
> Results for TextVQA
>
> Dataset | Model | Method | 1% | 5% | 10% | 20% | 40% | 60% | 80% | 100%
> --- | --- | --- | --- | --- | --- | --- | --- | --- | --- | ---
> ChartQA (Metric: Acc) | LLaVA-Mistral-7B | VL-Cache          | 34.7 | **59.1** | **64.4** | **65.4** | **65.4** | **65.4** | **65.4** | 65.4 |
> ChartQA (Metric: Acc) | LLaVA-Mistral-7B | H2O.                  | **43.0** | 56.5 | 63.4 | 63.4 | 62.4 | **65.4** | **65.4** | 65.4 |
> ChartQA (Metric: Acc) | LLaVA-Mistral-7B | PyramidKV         | 35.0 | 50.1 | 60.1 | 62.8 | **65.4** | **65.4** | **65.4** | 65.4 |
> ChartQA (Metric: Acc) | LLaVA-Mistral-7B | StreamingLLM   | 19.5 | 31.4 | 43.2 | 53.6 | 59.8 | 59.9 | 61.8 | 65.4 |
> ChartQA (Metric: Acc) | LLaVA-Mistral-7B | ZipCache          | 35.6 | 51.2 | 59.4 | 63.4 | **65.4** | 65.0 | **65.4** | 65.4 |
>
> Dataset | Model | Method | 1% | 5% | 10% | 20% | 40% | 60% | 80% | 100%
> --- | --- | --- | --- | --- | --- | --- | --- | --- | --- | ---
> ChartQA (Metric: Acc) | LLaVA-1.6-34B | VL-Cache | 12.9 | **70.7** | **71.4** | **73.4** | 72.4 | 73.6 | **72.7** | 72.7 |
> ChartQA (Metric: Acc) | LLaVA-1.6-34B | H2O | 18.3 | 47.0 | 63.2 | 69.5 | 71.5 | **74.4** | **72.7** | 72.7 |
> ChartQA (Metric: Acc) | LLaVA-1.6-34B | PyramidKV | **30.1** | 66.4 | 70.6 | 72.5 | 73.4 | 74.3 | **72.7** | 72.7 |
> ChartQA (Metric: Acc) | LLaVA-1.6-34B | StreamingLLM | 0.0 | 9.6 | 19.5 | 34.7 | 70.4 | 70.4 | 71.4 | 72.7 |
> ChartQA (Metric: Acc) | LLaVA-1.6-34B | ZipCache | 0.0 | 47.1 | 64.3 | 70.5 | **73.5** | 73.7 | **72.7** | 72.7 |
>
> Results for ChartQA
>
> Dataset | Model | Method | 1% | 5% | 10% | 20% | 40% | 60% | 80% | 100%
> --- | --- | --- | --- | --- | --- | --- | --- | --- | --- | ---
> TextVQA (Metric: Acc) | LLaVA-Mistral-7B | VL-Cache | 20 | **33** | **41** | 38 | **40** | **40** | **41** | 41 |
> TextVQA (Metric: Acc) | LLaVA-Mistral-7B | H2O | 23.0 | 26 | 32.0 | 39 | **41** | **40**| **41** | 41 |
> TextVQA (Metric: Acc) | LLaVA-Mistral-7B | PyramidKV | **27** | 29 | 39 | 39 | 38 | **40** | **41** | 41 |
> TextVQA (Metric: Acc) | LLaVA-Mistral-7B | StreamingLLM | 11 | 17 | 17 | 30 | 39 | 37 | 40 | 41 |
> TextVQA (Metric: Acc) | LLaVA-Mistral-7B | ZipCache | 23 | **33** | 40 | **42** | 40 | **40** | **41** | 41 |
>
> Dataset | Model | Method | 1% | 5% | 10% | 20% | 40% | 60% | 80% | 100%
> --- | --- | --- | --- | --- | --- | --- | --- | --- | --- | ---
> TextVQA (Metric: Acc) | LLaVA-1.6-34B | VL-Cache | 9 | 51 | **56** | 55 | **54** | **55** | **55** | 54 |
> TextVQA (Metric: Acc) | LLaVA-1.6-34B | H2O | 28 | 42 | 50 | 54 | **54** | 53 | 54 | 54 |
> TextVQA (Metric: Acc) | LLaVA-1.6-34B | PyramidKV | **31** | **52** | 54 | **56** | **54** | 54 | **55** | 54 |
> TextVQA (Metric: Acc) | LLaVA-1.6-34B | StreamingLLM | 5 | 7 | 11 | 14 | 43 | 51 | 52 | 54 |
> TextVQA (Metric: Acc) | LLaVA-1.6-34B | ZipCache | 0 | 43 | 55 | 52 | 54 | 54 | 54 | 54 |

---

> > ### Comment · Reviewer_XVzL · 2024-11-27
> >
> > Thanks to the authors for the response. I will raise my score as my concerns are addressed.

---

> > > ### Author Response · Authors · 2024-11-27
> > >
> > > Thank you Reviewer XVzL for letting us know that your concerns have been addressed and for raising your score!

---

### Official Review · Reviewer_XU5m · 2024-11-03

**Soundness:** 3
**Presentation:** 2
**Contribution:** 3
**Rating:** 6
**Confidence:** 4

**Summary:**

This paper addresses the challenge of efficiently storing and accessing large Key-Value (KV) caches in Vision-Language Models (VLMs), which are crucial for handling visual contexts like images or videos. Existing KV cache compression methods developed for Large Language Models (LLMs) don't perform optimally when applied to VLMs. To solve this, the authors propose VL-Cache, a compression method designed specifically for VLM inference. VL-Cache utilizes a layer-adaptive, sparsity-aware cache allocation and a modality-aware token scoring approach to reduce cache size without losing accuracy. Experimental results show that this method retains comparable accuracy with only 10% of the original cache, accelerates token generation by up to 2.33x, decoding by up to 7.08x, and reduces GPU memory usage by 90%.

**Strengths:**

1. The experimental results are very good; the performance is significantly better compared to other KV-cache methods.
2. Sparsity-aware layer-wise KV cache allocation and modality-aware token scoring are meaningful and have also been proven effective in the experiments.

**Weaknesses:**

1. The writing is somewhat complex and unclear; it takes a few careful readings to understand. Adding more illustrations to explain the method would improve it.
2. For MathVista, it seems that the KV-cache isn’t important? Even with only 1% of it, the performance is quite good, and sometimes the performance with 100% KV-cache isn’t even the best. Could you explain that?
3. In terms of writing, Section 3's PRELIMINARY EXPERIMENT is too long, while there’s relatively little written about the methodology. This writing style increases the burden on the reader. Readers likely want to see the method as soon as possible, so the focus should be on discussing your approach.

**Questions:**

Please see the section on weakness.

---

> ### Author Response · Authors · 2024-11-24
> **Response to Reviewer XU5m**
>
> We thank the reviewer for recognizing that our proposed method is well motivated and presents significantly better performance than existing methods. For concerns regarding the presentation of the paper, we address them as below.
>
>
> ### On Weakness 1
>
> We sincerely appreciate the reviewer’s time and efforts in evaluating our paper. To help readers understand our method more easily, we have significantly revised and updated Figure 4, which provides an overview of our proposed method. We have also added more transitions between sections to improve the flow of the paper.
>
> ### On Weakness 2
>
> These are great observations. On MathVista, multiple KV cache compression methods, including ours, perform well even with only 1% KV cache. One reason is that 54% of MathVista are multiple-choice questions. Selecting the correct answer is intuitively easier than generating it. This might explain the success of KV cache pruning methods even under extremely low cache budget. The other reason is that both H2O and our method essentially use full cache to generate the first token: H2O evicts tokens after prefill hidden states from a layer is computed, our method evicts tokens until prefill hidden states from the last layer is computed. Thus, in the extreme case of only one token is generated, both will recover full cache performance. Regarding why sometimes using partial KV cache outperforms full cache, H2O paper also observed this phenomenon, attributing it to the regularization effect of KV cache pruning.
>
> ### On Weakness 3
>
> The reviewer raised a valid concern over the presentation of our method. With the current structure, we intended to use Section 3 to motivate our method in Section 4 so that readers can understand where we come from more easily. However, we do realize this could distract the readers, so we have trimmed Section 3 so that our method is introduced sooner. We have also made several improvements to enhance the presentation of Section 4, such as improving Figure 4 and adding detailed explanations to our proposed algorithm.

---

> > ### Comment · Reviewer_XU5m · 2024-12-02
> >
> > I would like to thank the authors for their reply and their answers to my questions. I apprechiate that they were incorporated into the main text and I maintain my score.

---

### Author Response · Authors · 2024-11-24
**Rebuttal by Authors**

We would like to thank all reviewers for their time and effort in reviewing our work. We are glad that our work receives the following recognition from our reviewers:

1. In terms of motivation and methodology: “the insight that the attention patterns of generated tokens closely align with language tokens is valuable” (XVzL); the observation that “KV cache budget and compression ratios differ in layers and modalities” provides “good insights into the optimization of VLM cache compression” (KiPr); “the sparsity analysis of prefilling and decoding effectively motivates the use of adaptive sparsity” (X8F2);“sparsity-aware layer-wise KV cache allocation and modality-aware token scoring are meaningful and have also been proven effective in the experiments” (XU5m).
2. In terms of experimental results: “the experimental results are very good; the performance is significantly better compared to other KV-cache methods” (XU5m); “the experimental results show that the proposed methods achieve significant inference speedups and GPU memory savings” (KiPr); “the proposed method achieves significant speedup in decoding” (X8F2).

We show additional experimental results requested by multiple reviewers in this reply and have addressed other concerns in individual threads. We have also uploaded an updated version of our paper for further reference.

Both Reviewer XVzL and X8F2 have requested comparisons of our method with ZipCache’s scoring policy, which uses normalized attention scores to evict tokens. We show the results in the table below. We found that our method presents consistently better performance with 10% KV cache. In our updated paper, we also show that using post vision attention (our method) leads to higher cache hit rate as well. Interestingly, normalized attention scores proposed in ZipCache do not consistently improve upon accumulated attention scores used in H2O as shown in the ZipCache paper when tested on vision-language tasks, which highlights the importance of specialized scoring policies as we proposed in VL-Cache for VLMs.

Dataset | Model | Method | 1% | 5% | 10% | 20% | 40% | 60% | 80% | 100%
--- | --- | --- | --- | --- | --- | --- | --- | --- | --- | ---
Coco-Caption (Metric: CIDEr) | LLaVA-Mistral-7B | VL-Cache | 2.64 | **82.53** | **100.36** | **102.06** | **99.93** | **101.07** | 100.08 | 100.68
Coco-Caption (Metric: CIDEr) | LLaVA-Mistral-7B | ZipCache | **4.4** | 53.00 | 66.41 | 92.36 | 99.51 | 100.71 | **102.86** | 100.68
Coco-Caption (Metric: CIDEr) | LLaVA-1.6-34B | VL-Cache | 0.00 | **120.11** | **137.35** | **139.42** | 138.58 | 139.19 | **138.01** | 135.07
Coco-Caption (Metric: CIDEr) | LLaVA-1.6-34B | ZipCache | 0.00 | 12.92 | 131.61 | 134.98 | **140.59** | **140.63** | 137.90 | 135.07
DocVQA (Metric: ANLS) | LLaVA-Mistral-7B | VL-Cache | **43** | **59** | **62** | **64** | **67** | 67 | **67** | 68
DocVQA (Metric: ANLS) | LLaVA-Mistral-7B | ZipCache | 41 | 40 | 60 | 45 | 65 | **68** | **67** | 68
DocVQA (Metric: ANLS) | LLaVA-1.6-34B | VL-Cache | **41** | **82** | **84** | **85** | **85** | **85** | **85** | 85
DocVQA (Metric: ANLS)| LLaVA-1.6-34B | ZipCache | 0 | 55 | 74 | 81 | **85** | **85** | **85** | 85
MathVista (Metric: ACC) | LLaVA-Mistral-7B | VL-Cache | **38** | **36** | **39** | **40** | **40** | **42** | **42** | 41
MathVista (Metric: ACC) | LLaVA-Mistral-7B | ZipCache | 35 | 35 | **39** | 34 | 40 | 42 | 40 | 41
MathVista (Metric: ACC) | LLaVA-1.6-34B | VL-Cache | **41** | **42** | **42** | **44** | **45** | **45** | 42 | 43
MathVista (Metric: ACC) | LLaVA-1.6-34B | ZipCache | 28 | 35 | 41 | 41 | 42 | 43 | **43** | 43

---

### Author Response · Authors · 2024-11-27
**Thank-You Note**

To all reviewers,

Thanks again for taking the time to review our paper and rebuttal. Your constructive feedbacks have helped us greatly in improving the quality of our submission, and we really appreciate that!


Best,

Authors

---

### Meta-Review · Area_Chair_KLR4 · 2024-12-18

**Metareview:**

The paper introduces VL-Cache, a novel KV cache compression method tailored for accelerating vision-language model inference, which significantly reduces memory usage and latency while maintaining high accuracy.

After rebuttal and discussions, this paper receives all positive ratings. After carefully  reviewing the paper and all the reviewers' comments and discussions, The AC agrees to accept the paper and strongly recommends incorporating the content from the rebuttal into the final version.

**Additional Comments On Reviewer Discussion:**

- Reviewer XU5m acknowledges the good experimental results and the meaningfulness of the proposed methodologies, such as sparsity-aware layer-wise KV cache allocation and modality-aware token scoring, but suggests improvements in presentation and clarity, including the addition of more illustrations and a more balanced presentation of methodology and results.

- Reviewer XVzL recognizes the value of the insight that attention patterns of generated tokens align closely with language tokens and appreciates the novelty of the dynamic cache budget allocation mechanism and post-vision language tokens usage. However, concerns are raised about the writing clarity, the importance of KV cache for certain tasks, and the need for more challenging benchmarks.

- Reviewer KiPr commends the paper for addressing the I/O bottleneck in VLMs and providing good insights into KV cache optimization. However, the reviewer points out the need for clearer methodology explanations, a better definition of Cache Hit Rate, enhanced clarity in figures, and additional experimental details such as I/O traffic amounts.

- Reviewer X8F2 praises the paper's well-written and easy-to-follow nature, the effective motivation from sparsity analysis, and the significant decoding speedups achieved. The reviewer also requests clarifications on notation, the actual impact of KV cache compression on memory usage, detailed breakdowns of prefill and decoding times, and additional comparisons with other VLM acceleration methods.

During rebuttal and discussions, the authors have addressed the reviewers' concerns by revising the paper for clarity, providing additional experimental results, and comparing their method with other state-of-the-art approaches. Most of the previous concerns are properly addressed and two reviewers raise the scores. The final consensus of positive ratings lead to a acceptance for this submission.

---

### Decision · Program_Chairs · 2025-01-22

Accept (Poster)